# How to Sustain Fisheries: Expert Knowledge from 34 Nations

**Jessica A. Nilsson** [1,2,*], **Elizabeth A. Fulton** [1,3], **Craig R. Johnson** [2] **and Marcus Haward** [2,3]

1    CSIRO Oceans & Atmosphere, GPO Box 1538, Hobart, Tasmania 7001, Australia; beth.fulton@csiro.au
2    Institute for Marine and Antarctic Studies, University of Tasmania, Private Bag 129, Hobart, TAS 7001, Australia; craig.johnson@utas.edu.au (C.R.J.); marcus.haward@utas.edu.au (M.H.)
3    Centre for Marine Socioecology, University of Tasmania, Hobart, TAS 7001, Australia
*    Correspondence: jas.nilsson@gmail.com; Tel.: +46-765386215

**Abstract:** Ensuring productive and sustainable fisheries involves understanding the complex interactions between biology, environment, politics, management and governance. Fisheries are faced with a range of challenges, and without robust and careful management in place, levels of anthropogenic disturbance on ecosystems and fisheries are likely to have a continuous negative impact on biodiversity and fish stocks worldwide. Fisheries management agencies, therefore, need to be both efficient and effective in working towards long-term sustainable ecosystems and fisheries, while also being resilient to political and socioeconomic pressures. Marine governance, i.e., the processes of developing and implementing decisions over fisheries, often has to account for socioeconomic issues (such as unemployment and business developments) when they attract political attention and resources. This paper addresses the challenges of (1) identifying the main issues in attempting to ensure the sustainability of fisheries, and (2) how to bridge the gap between scientific knowledge and governance of marine systems. Utilising data gained from a survey of marine experts from 34 nations, we found that the main challenges perceived by fisheries experts were overfishing, habitat destruction, climate change and a lack of political will. Measures suggested to address these challenges did not demand any radical change, but included extant approaches, including ecosystem-based fisheries management with particular attention to closures, gear restrictions, use of individual transferable quotas (ITQs) and improved compliance, monitoring and control.

**Keywords:** ocean governance; fisheries management; ecosystem-based management; overfishing; sustainable fishing

## 1. Introduction

For the second half of the twentieth century, scientific and technological endeavours focused on finding new fisheries to exploit and more efficient and effective ways of harvesting. This was possible as developments in vessel and gear design, navigation and positioning systems and means to detect fish (e.g., depth-sounders) became more accessible to the common fisher [1]. These scientific and technological advances led to a dramatic increase in global fishing effort. Such developments also allowed fleets to exploit more distant resources to the point where the only unexploited fishery resources were those that remained physically inaccessible, for example under sea-ice [2]. For much of this period, much of the sea was treated as a common resource with many fish stocks exploited with little restriction and only a few with strict governance, setting conditions for a "tragedy of the commons" [3]. In recent decades, there has been increasing awareness of the need for global political action on natural resource management, as evidenced by the Rio Declaration on Environment

and Development in 1992 [4] and by such initiatives as the Oxford Martin Commission for Future Generations, launched in 2012 by an interdisciplinary group of organisations [5].

By the latter decades of the twentieth century, it became apparent that the substantial increase in fishing capacity was leading to overexploitation and, in some cases, collapse of fisheries [6,7]. Overfishing, with associated ecosystem shifts, is a major threat to the marine environment. More than half of the world's marine fish stocks are considered to be either overexploited or fully exploited with no room for further expansion [8]. Although stocks have been fished for a number of centuries, the sheer number of global stocks that are currently below sustainable exploitation levels is unprecedented [8,9]. Failure to understand and sustain ecosystem processes, including human impacts upon them, continues to cause major biodiversity loss in many places around the globe [10–14]. As a result, a number of scientific initiatives are directed towards developing and applying methods to better measure, predict and monitor sustainable yields of key fish stocks, in both national and international waters [15,16].

## 1.1. Public Demand for Marine Management

Over at least two decades, there have been increasing calls from scientists, nongovernmental organisations (NGOs) and the public at large for better management of marine ecosystems. These calls have partly been based on scientific research that has revealed the myriad ways that fishing activities (along with climate change, terrestrial runoff and other anthropogenic processes) impact the overall health of marine ecosystems [9,17,18]. Increased environmental awareness has led to calls for attention to ecosystem-focused approaches to management, variously termed the Ecosystem Approach to Fisheries (EAF) [8], Ecosystem-Based Fisheries Management (EBFM) [19], or cross-sectoral Ecosystem-Based Management (EBM) (i.e., spanning all marine sectors, not just fisheries) [20].

Despite an increase in scientific knowledge and management efforts on overexploited fisheries and marine systems, there are still ecosystems and fish stocks showing no or little sign of recovery. It is recognized that impacts on the marine environment from fishing pressure might, in some cases, be more severe than first thought [21]. This calls for fisheries to be governed and managed holistically, needing a combination of environmental, biological and socioeconomic research to provide robust marine governance and management strategies to ensure a sustainable marine environment. The gap, however, between science and policy has been acknowledged [22,23], as has the fact that governance and management decisions are not always based on the best science available [24].

## 1.2. The Management Challenge: Predicting Uncertainties

Apart from fishing pressure, marine ecosystems and fisheries are also subject to other effects of human activity, such as climate change, ocean acidification and related biophysical impacts, habitat loss and impacts from terrestrial land use, such as land-based sources of pollution and litter [12,25,26]. A key challenge is to predict the long-term effects of these cumulative anthropogenic impacts and to form appropriate management strategies [27]. Without appropriate knowledge and understanding of the ecosystem supporting fisheries, and the communities in which fisheries are embedded, it is likely that management will fail [28].

The complexity of governing and managing fisheries in a socioeconomic context was illustrated by the 2009 Nobel Prize in Economics. The Nobel Prize was shared between Dr Ostrom, whose research was based on the assumption that people in a community can create successful agreements (and compliance) for managing common use of natural resources, such as fisheries [29], and by Dr Williamson, who presumed that natural resource management needs a top-down management approach because individuals ultimately cannot trust one another [30].

Another challenge (at times the largest challenge) for fisheries and environmental managers is a lack of political will to use and implement recommendations based on scientific findings. This challenge can reflect and reinforce the 'science–policy gap' [22]. Although scientists may make management recommendations based on their findings, ultimately management decisions are made by government officials and politicians. Importantly, these decisions are not driven only by scientific knowledge of the

stock and dynamics of the ecosystem in which a fishery is embedded, but also by a range of political agendas and economic, social and cultural considerations. While scientists may be frustrated with this reality, it is important for them both to accept that they are only one voice at the decision-maker's table, but also not to shy away from objectively presenting the scientific evidence.

Given that there are many environmental, biological and socioeconomic factors that ultimately affect the state and health of the oceans, and that these drivers vary in time and space, decision-makers increasingly ask whether there is sufficient scientific information and knowledge of ecological functions and processes to implement an ecosystem approach to marine and fisheries management [31]. Successful marine management needs careful integration across sound scientific knowledge, development and implementation of management instruments and compliance tools. Even though there are many ecological processes to understand further, it is widely recognised that we do have sufficient scientific information to start implementing EBFM in many places around the world [32–34].

One challenge to implementing EBFM is that ocean resources are often managed sector-by-sector, i.e., coastal and terrestrial development, water management, environment conservation and primary industries (including fisheries) are each managed by separate jurisdictions [31]. The different set of goals and objectives within each sector may have implicit trade-offs so that fisheries managers often need to navigate and respond to conflicting objectives and incentives involving two or more government agencies [35,36] or interest groups. Clearly, if there is a negative impact on marine habitat due to fishing gear as well as from toxic terrestrial run-off, then both the fishing sector and the land-use sector need to take appropriate actions to prevent further habitat degradation [37]. Implementing EBFM, or EBM, requires a governmental organisational structure that matches this holistic view of ecosystem-based management. This does not immediately dictate an overarching, all-encompassing regulatory body, but it does necessitate communication (and where possible harmonisation of requirements) between agencies.

While defining the final scope of an ecosystem-based management governance system is beyond the scope of this paper, providing information on the current state of play is important to understanding what steps are still required to achieve solid advances. This research explores the main issues influencing the sustainability of fisheries. It draws on data derived from an international survey of fisheries experts, using the elicited responses to (1) identify the main issues in attempting to ensure the sustainability of fisheries, and (2) address how to begin to bridge the gap between scientific knowledge and the governance of marine systems, from the point of view of fishery management experts. The survey data were analysed to explore expert insights, opinion and understanding on the challenges to sustainable fisheries, the efficacy of tools used to manage fisheries and the complexity of interactions in fishery socioecological systems.

## 2. Methods

### 2.1. Data Collection

We targeted marine experts from around the world, primarily scientists and natural resource managers. Our survey was designed to elicit knowledge from marine scientists, managers, fishers and policy-makers. The intention was to gather specialist knowledge and experience in relation to sustaining fisheries. The survey was implemented by inviting experts to share their knowledge and experiences at the 6th World Fisheries Congress in Edinburgh, 8–11 May 2012. Attendees were invited to sit down at a booth and take part in the web-based survey. If an individual did not have time to conduct the survey when approached, they were given the opportunity to complete the survey in their own time either online or via a hard-copy of the survey. In total, 549 persons were invited to participate in the survey, resulting in 168 fully completed surveys (20 more provided partial completions that were still sufficient for inclusion in the analysis), giving a 34% response rate.

*2.2. Analysis*

The questions and a summary of the answers are presented in Appendix A. Given small sample sizes when respondents were broken down by category, for some questions, the responses from fisheries/natural resource managers and policy-makers were aggregated into a 'managers/policy makers' group. For the same reason, variables measured on five-point response scales were, in some cases, converted into a three-point scale. For example, the five-point 'satisfied-dissatisfied' scale was in some cases collapsed into the categories 'satisfied', 'neutral' and 'dissatisfied', by combining 'satisfied' with 'very satisfied', and 'dissatisfied' with 'very dissatisfied'.

Statistical analyses, including crosstabulations, were conducted using SPSS (Version 25.0., IBM Corp, Armonk, NY, USA). No corrections were made. The statistical independence of pairs of variables was analysed using the 2-factor G-test for independence at a 95% significance level.

## 3. Results

*3.1. Demographics*

The respondents were from 34 nations, representing scientists, fisheries managers, fishers, policy-makers, NGOs and others. Forty (40) respondents were from Australia, as the survey was trialed there before presenting it at the World Fisheries Congress.

Seventy-one percent of the respondents were male, and 60% of the respondents were 35–64 years old (Appendix A). Forty-two percent of the respondents had a Doctoral degree, 28% a Master's degree, 14% a 3–4 year university degree, and the remainder did not hold a degree, but all had completed high school (Appendix A). The majority of the respondents were scientists (Figure 1), with fifty-nine percent of the respondents holding a degree in marine science and 20% in environmental science. Other respondents had degrees in business, law, economics and social sciences (Appendix A).

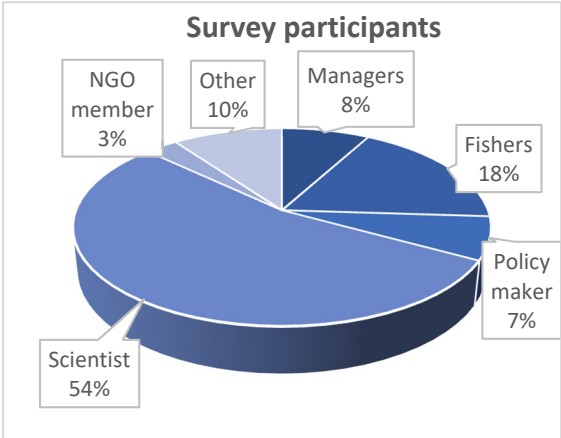

**Figure 1.** The breakdown of respondents by profession (n = 177). 'Other' includes consultants, economists, social scientists, lawyers and students. NGO, nongovernmental organization.

The majority of the respondents spanned middle-executive management positions, and represented pelagic, demersal, coastal and crustacean fisheries (Figures 2 and 3). The respondents represent experience and knowledge from fisheries deemed to be sustainable as well as from overfished, collapsed, recovering and exploratory fisheries (Figure 4). Of the respondents, 47% worked with national management agencies, 24% with international management and 15% at universities (Appendix A).

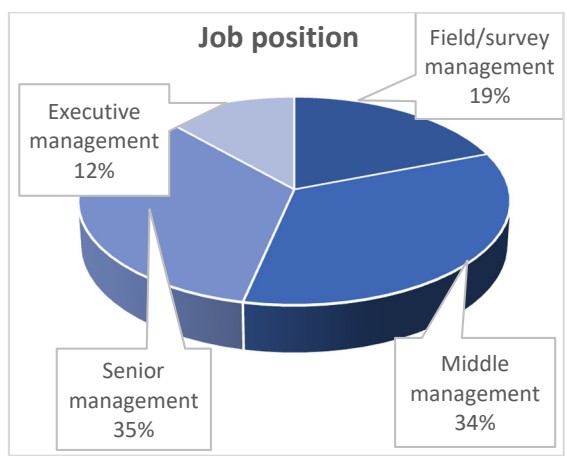

**Figure 2.** The job position held by respondents (n = 146).

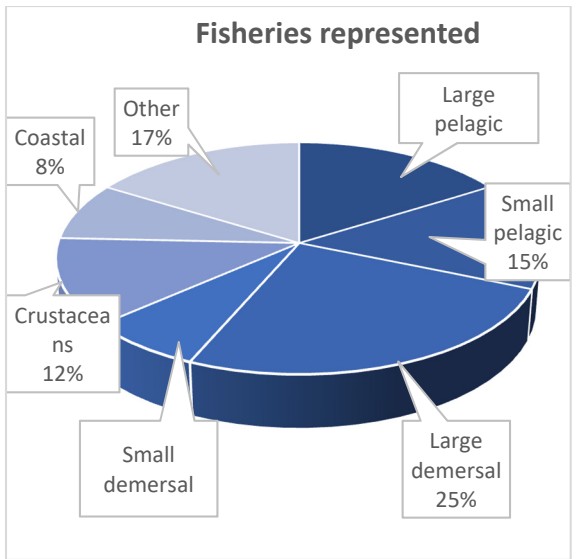

**Figure 3.** The fishery types covered by survey respondents. 'Other' includes shark, inland, aquaculture and shellfish (n = 143).

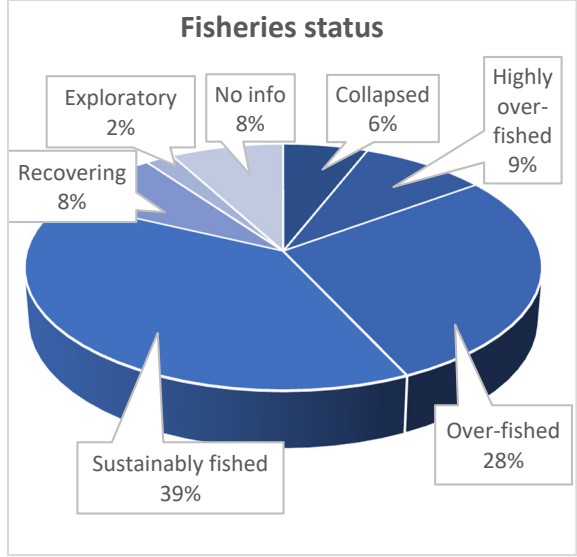

**Figure 4.** The status of the fisheries the respondents are working with (n = 172).

### 3.2. Anthropogenic Effects on Fisheries and Marine Systems

Overfishing, climate change and habitat destruction were believed to be the three threats most affecting fisheries, both at national and global scales (Figure 5). There was no significant difference among the responding groups as to whether or not they perceived the same 10 threats as major threats to national and world fisheries (G = 10.191, df = 9, p = 0.335), where G is the likelihood-ratio, df the degree of freedom and p the probability value.

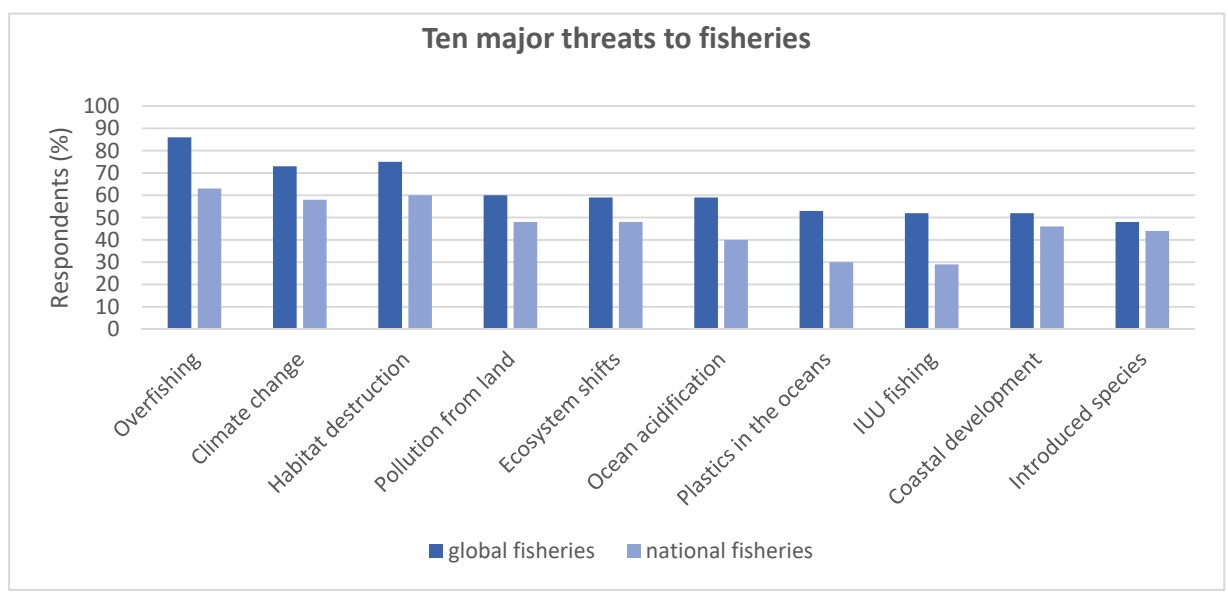

**Figure 5.** The 10 major threats to national and global fisheries (n = 164).

Overfishing was believed to be a major threat to world fisheries by 79% of the managers, 92% of the policy-makers, 79% of the scientists and 84% of the fishers (Figure 5). Notably, 69% of the policy-makers and scientists said they believe that illegal, unreported and unregulated (IUU) fishing is not a major threat to national fisheries, while 78% of the fishers said they think it is.

Fifty-eight percent of all respondents believed climate change to be a major threat to national fisheries, while 59% believed that ocean acidification is a major threat to world fisheries and 40% to national fisheries. Seventy-two percent of the fishers said they think habitat destruction is a major threat to the marine environment for world fisheries, while only 13% said it is a threat to national fisheries. Forty-one percent of the scientists believed land-based pollution is a major threat to fisheries, compared to 84% of the fishers, 85% of the policy-makers and 79% of the managers. Of all the respondents, 46% said plastic is a major threat to world fisheries (57% of managers and 62% of the scientists) and 30% said it is a major threat to national fisheries.

Despite the divergence in views in the earlier question pertaining to whether IUU is a threat to international or national fisheries, there was no significant difference among the responding groups on how they viewed the specific aspects of IUU fishing (G = 61.275, df = 45, p = 0.054). Corruption was seen as the main aspect of IUU fishing (66%), with 55% of respondents believing that there is insufficient compliance in place to combat IUU fishing (Figure 6). Sixty-four percent said they believe IUU fishing is a problem within their fishery, and of those 43% said they think IUU fishing amounts to 6–30% of the total catch (Appendix A). When specifically asked about IUU (rather than ranking it against other threats), on a global scale, 99% of the respondents believed that IUU fishing is a problem and 65% estimated the global level of IUU fishing to be between 31–60% of the total catch worldwide (Appendix A).

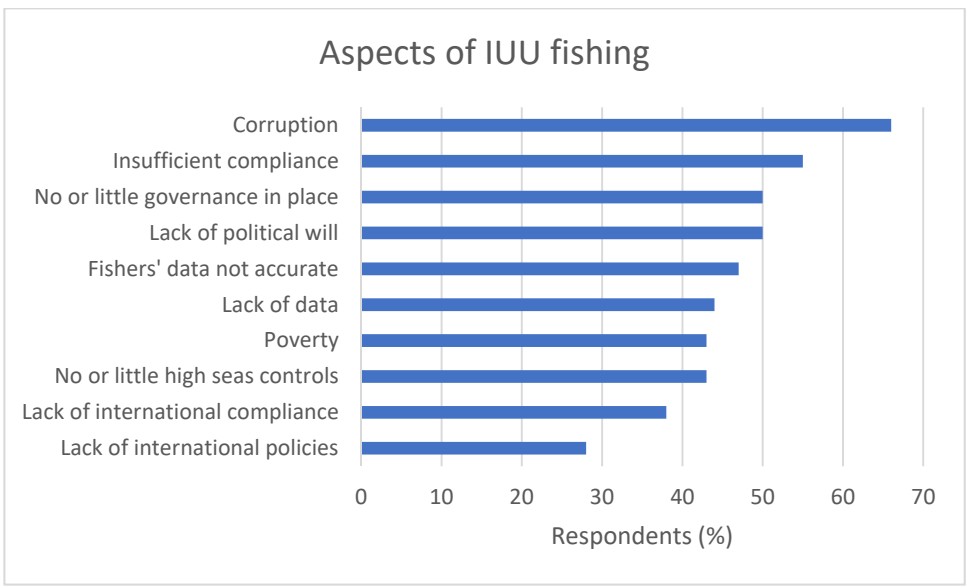

**Figure 6.** Key aspects of illegal, unreported and unregulated (IUU) problems identified by the respondents.

*3.3. Fisheries Governance and Management Affecting Fisheries and Marine Systems*

On the question of what the three main challenges to fisheries are, the following four factors ranked the highest: a lack of political will (56%); not enough compliance with regulations (33%); overfishing (29%); and stock assessment and monitoring (28%) (Figure 7). There was no significant difference among the responding groups regarding which of the four factors were seen as the main challenges to managing fisheries (G = 23.409, df = 15, p = 0.076). Despite compliance being listed as a major challenge to sustainability, 90% of the fishers and 66% of the scientists said there is already enough compliance.

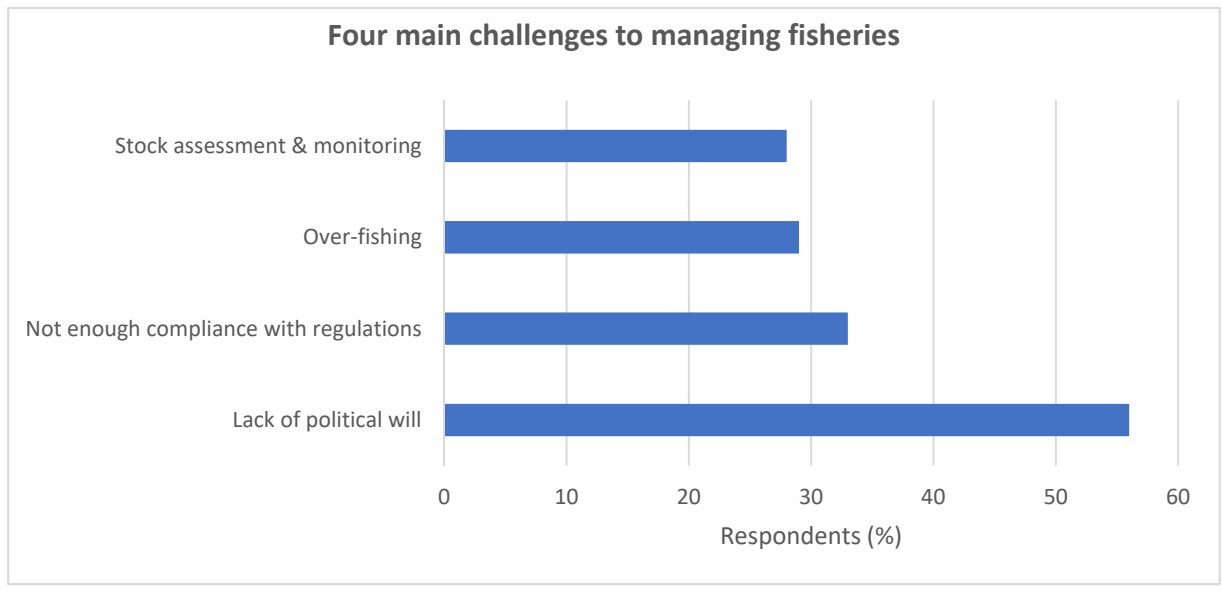

**Figure 7.** Expert opinions on four main challenges to managing fisheries (n = 174).

Fifty-five percent of the respondents believed that, during the course of their careers, they have seen major changes in fisheries management, such as increased input from scientists and industry, and stakeholder collaboration (Figure 8).

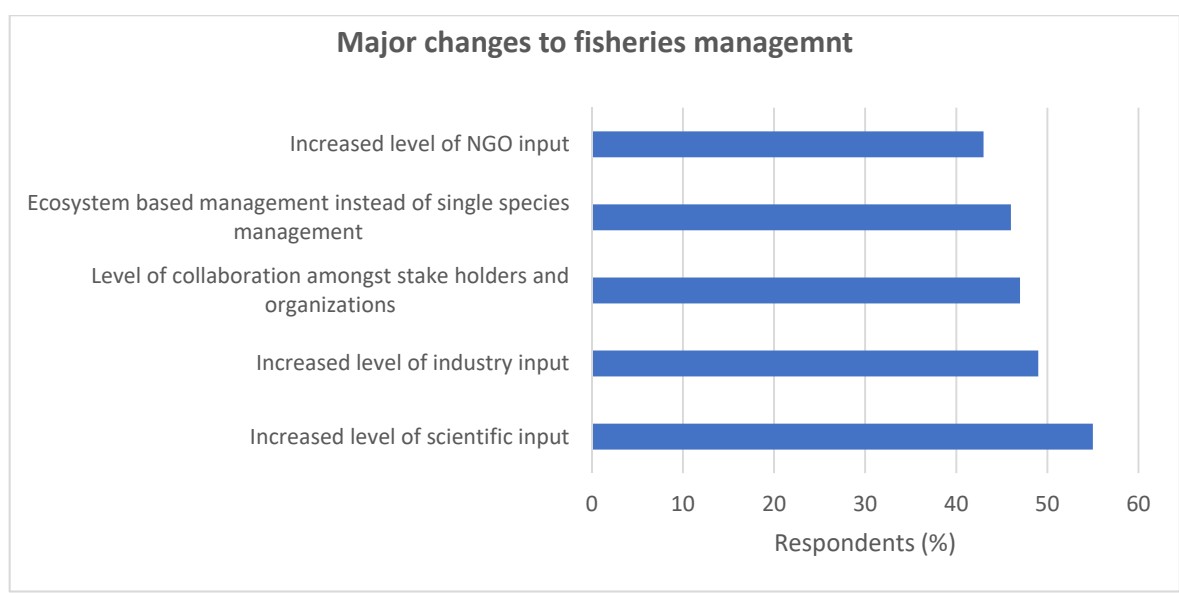

**Figure 8.** Major changes that have occurred in fisheries management during the respondents' careers in fisheries (n = 109).

More of the respondents were satisfied than dissatisfied with the planning and implementation of the EBFM processes. However, when considering the results of EBFM, a greater number of respondents were neutral, out numbering those who were satisfied or dissatisfied (Figure 9). When looking to the fisheries they knew best, 60% of the respondents said that the fishery they worked with has implemented (EBFM) (Appendix A), or a similar holistic approach to governing fisheries, though 50% said they were unsure as to whether the implementation of EBFM has been successful (Figure 10).

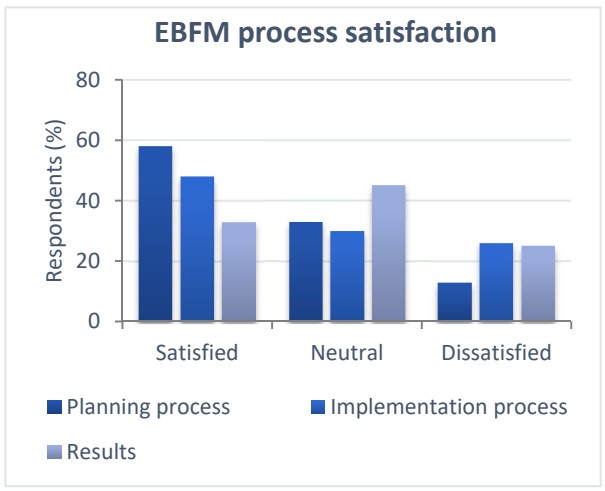

**Figure 9.** Measuring how satisfied the respondents were with the whole Ecosystem-Based Fisheries Management (EBFM) process (n = 104).

There was no significant difference among the responding groups in terms of their satisfaction with the planning processes associated with implementing EBFM (G = 11.358, df = 10, p = 0.33), with 73% of the managers, 67% of the policy-makers, 47% of the scientists and 50% of the fishers being satisfied. Thirty-eight percent of the scientists and 50% of the fishers were neutral. When it came to taking the step of implementing EBFM, there was also no significant differences among the responding groups on how they felt regarding this implementation process (G = 21.174, df = 15, p = 0.131), with approximately 50% of both the scientists and fishers being neutral.

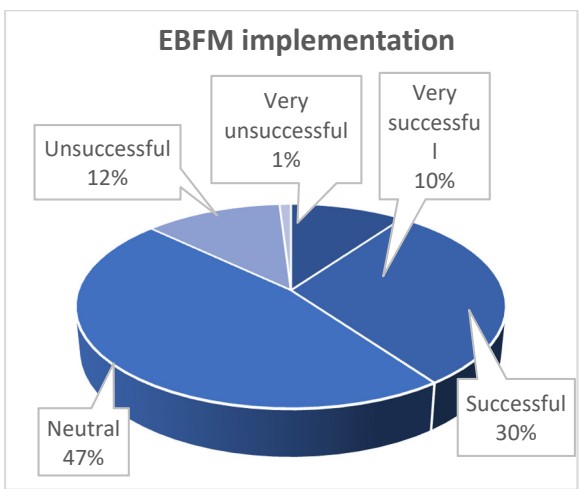

**Figure 10.** The perception of those respondents who said the EBFM process has been implemented regarding how successful the process had been (n = 107).

Sixty-four percent of the managers and 58% of the policy-makers were satisfied with the results of implementing EBFM, compared with 31% of the scientists, 46% of the fishers and 0% of the NGOs (Table 1). About as many scientists as managers thought the implementation process of EBFM had been unsuccessful (Table 1) and about as many fishers as scientists remained neutral as to whether the EBFM implementation process had been successful (Table 1).

**Table 1.** The level of success for the implementation process of EBFM per responding group (% within each responding group. n = 108).

|                   | Managers | Policy-Makers | Scientists | Fishers | NGOs |
|-------------------|----------|---------------|------------|---------|------|
| Very successful   | 0%       | 15%           | 11%        | 11%     | 0%   |
| Successful        | 64%      | 31%           | 20%        | 35%     | 0%   |
| Neutral           | 18%      | 39%           | 50%        | 54%     | 67%  |
| Unsuccessful      | 9%       | 15%           | 19%        | 0%      | 33%  |
| Very unsuccessful | 9%       | 0%            | 0%         | 0%      | 0%   |

Once EBFM is in place (often in an adaptive management context), it is important to know if it is proving successful. When asked about this, there was no significant difference among the responding groups regarding how satisfied they were with the results of EBFM (G = 16.571, df = 10, p = 0.084): 55% of the managers were satisfied, compared with 23% of the scientists (Table 2). Of the fishers, 65% were neutral and 67% of the NGOs were dissatisfied (Table 2). Figure 11 shows that EBFM is challenging to implement, mainly because the process is highly complex.

**Table 2.** Satisfaction among the responding groups regarding results of the implementation of EBFM (% within each responding group. n = 104).

|                   | Managers | Policy-Makers | Scientists | Fishers | NGOs |
|-------------------|----------|---------------|------------|---------|------|
| Very satisfied    | 0%       | 25%           | 2%         | 8%      | 0%   |
| Satisfied         | 55%      | 17%           | 21%        | 23%     | 33%  |
| Neutral           | 27%      | 33%           | 41%        | 65%     | 0%   |
| Dissatisfied      | 9%       | 25%           | 29%        | 4%      | 67%  |
| Very dissatisfied | 9%       | 0%            | 7%         | 0%      | 0%   |

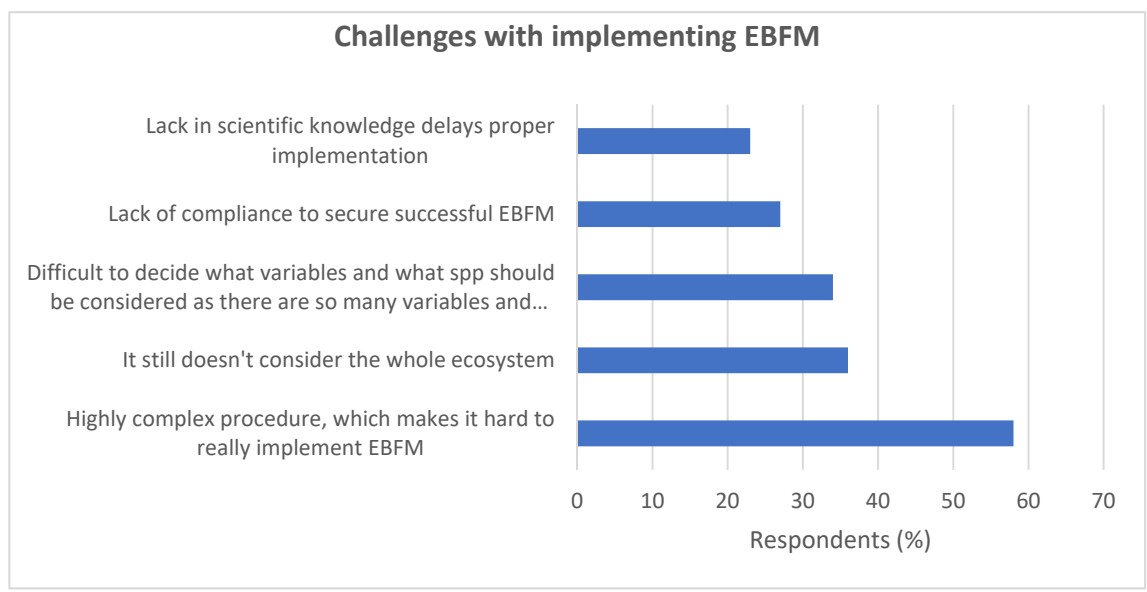

**Figure 11.** Implementing EBFM is a complex task (n = 83).

There was a significant difference among the responding groups regarding which tools are most efficient for implementing EBFM (G = 44.226, df = 20, p = 0.001). Respondents viewed good science, Marine Protected Areas (MPAs), individual transferable quotas (ITQs), gear restrictions and stakeholder participation to be the five most efficient tools for Ecosystem-Based Fisheries Management (Figure 12).

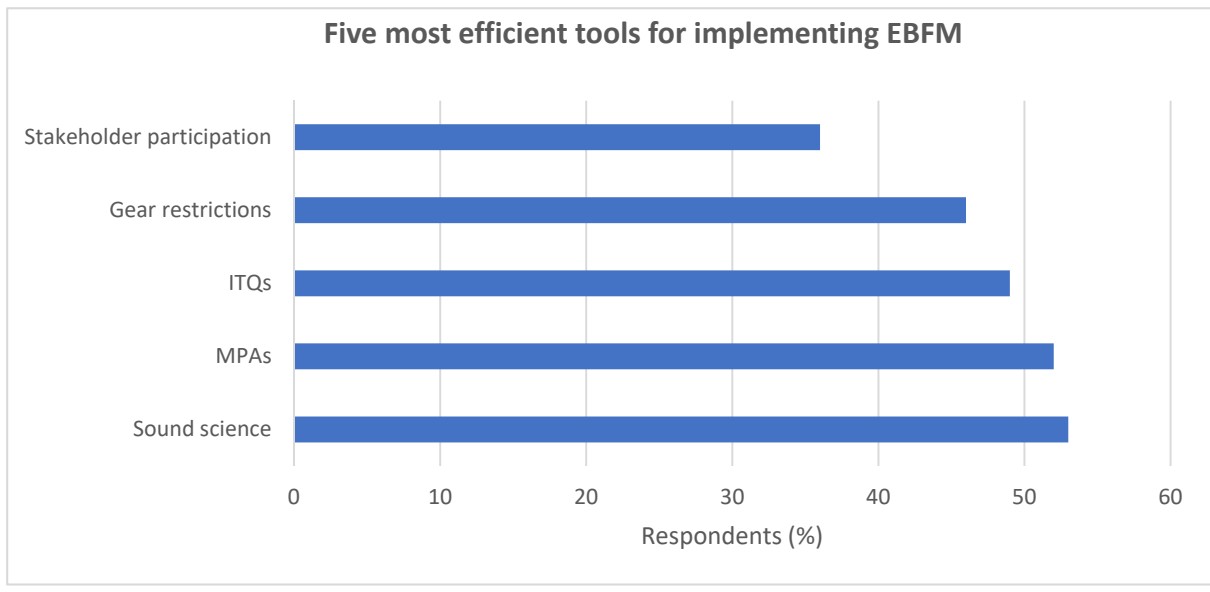

**Figure 12.** Participants' responses to the five most-efficient regulations for Ecosystem-Based Fisheries Management (n = 121). ITQs, individual transferable quotas.

### 3.4. Improvements Needed to Obtain and Maintain Sustainable Fisheries

For the question on what type of organisation would be optimal for implementing EBFM, 83% believed that a mix of a top-down and bottom-up management is optimal (Appendix A). When it came to what more is needed to sustain fisheries, 72% of all respondents answered they believe a stronger political will is needed to achieve successful ecosystem-based management (Figure 13).

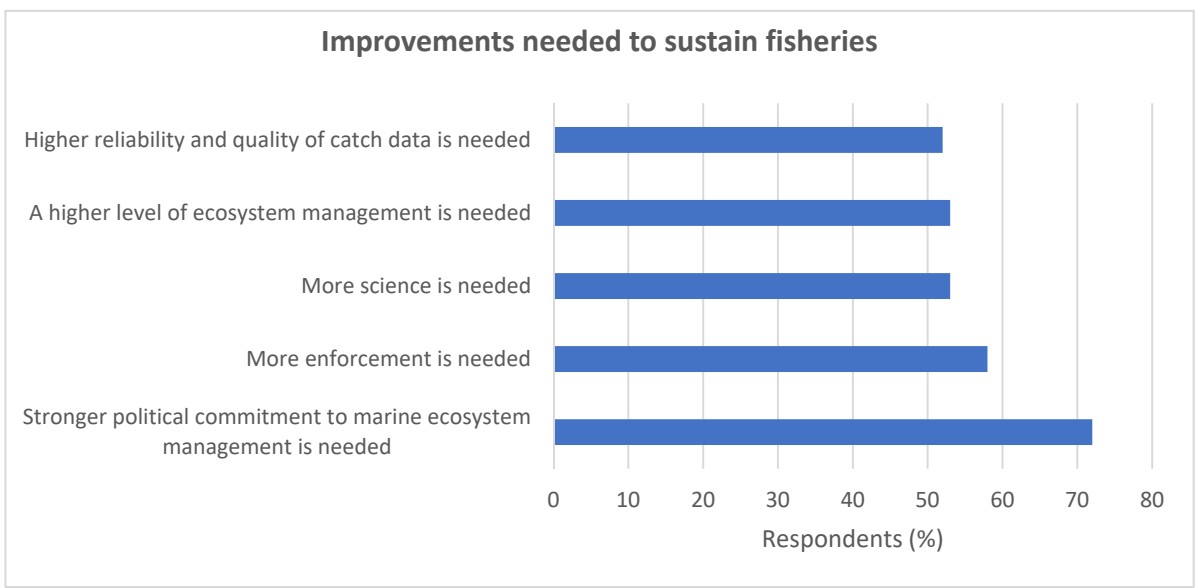

**Figure 13.** Improvements needed to obtain/maintain sustainable fisheries (n = 165).

There was no significant difference among the responding groups regarding which improvements are needed to sustain fisheries (G = 5.747, df = 20, p = 0.999), with all groups identifying the same mix of factors. However, this congruence did hide some differences in detail. Amongst managers, a clear majority (79%) stated that stronger political will is needed. A majority of managers (60%) also said they think more enforcement is needed; this latter result is in sharp contrast to the 25% of fishers who felt the same way. Overall, 53% of the respondents believed that more science is needed in order to obtain and maintain sustainable fisheries (Figure 13).

The majority of the respondents were supportive of input controls, such as by-catch reduction devices, size limits, spawning and spatial closures, regional zoning, seasonal closures and gear restrictions (Figure 14). The majority of the respondents also showed support for output controls, such as total allowable catch (86%), individual transferable catch (69%) and bag limits (69%) (Appendix A).

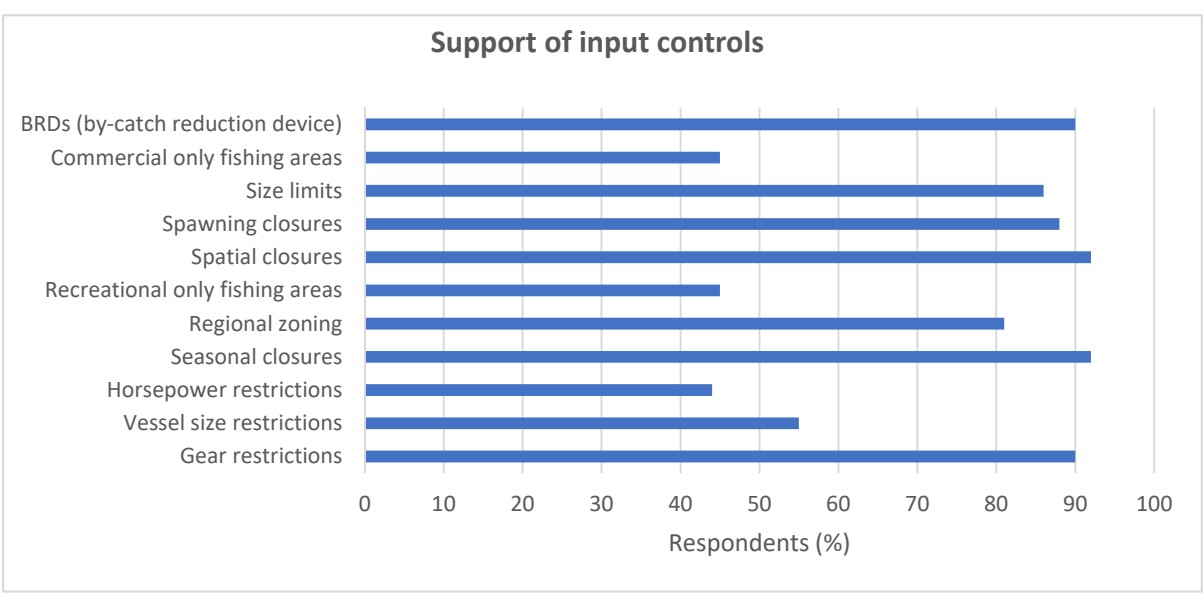

**Figure 14.** The level of support for several input controls shown by marine experts (n = 162).

When it came to monitoring and assessing stocks, Catch Per Unit Effort (CPUE) was the most common method used for measuring fish abundance (Figure 15), although logbook data was considered a close second.

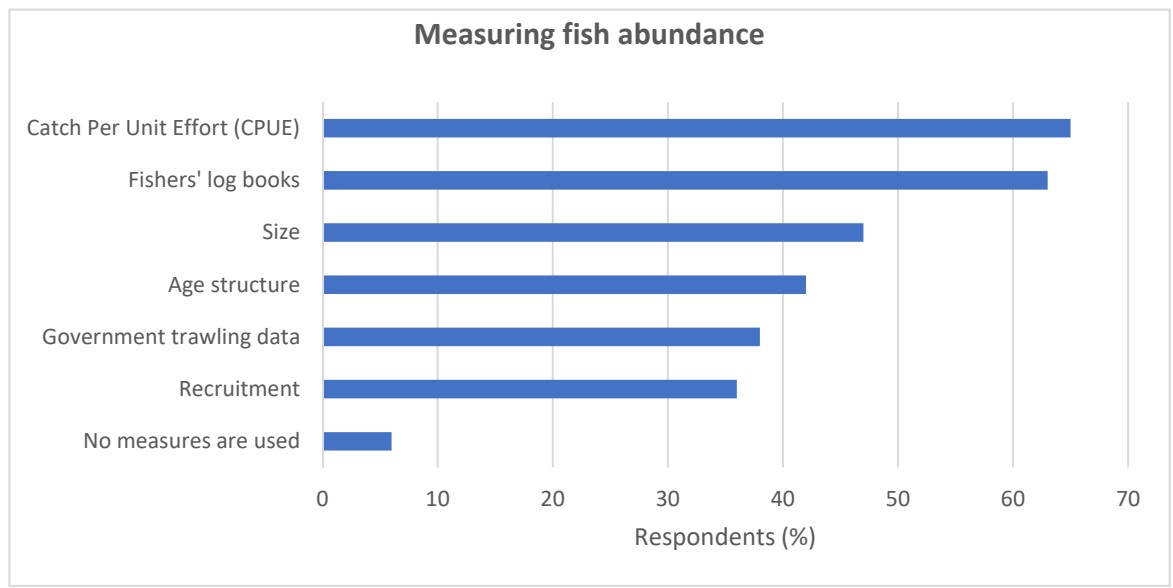

**Figure 15.** The prevalence of different approaches to measuring fish abundance.

Experts were asked to identify what they see as the main challenges to sustainable fisheries and what management tools would be generally useful for combatting challenges in fisheries (Table 3). Interestingly, while the challenges included things that are beyond the scope of fisheries management alone (e.g., land-based pollution or plastics), all of the suggested tools are classical fisheries management tools. When asked the question regarding why regulated fisheries are still faced with overexploitation, the highest ranking responses were: (1) the need for more scientific information; (2) existing science not being used to its fullest; and (3) a lack of political will. There was no significant difference to these three reasons among the responding groups (G = 2.001, df = 10, p = 0.996). The vast majority of all responding groups (regardless of background) said that the lack of political will is a major reason why regulated fisheries are still faced with overexploitation (Table 4).

**Table 3.** Ten main challenges and ten main tools for sustaining fisheries (n = 133).

| Ten Fisheries Challenges | Ten Tools for Sustain Fisheries |
|---|---|
| Overfishing | Seasonal closures |
| Climate change | Total Allowable Catch (TAC) |
| Habitat destruction | Size limits |
| Pollution from land | Spatial closures (e.g., MPA) |
| Ecosystem shift | Ecosystem-Based Fisheries Management (EBFM) |
| Ocean acidification | Spawning closures |
| Plastics in the oceans | Mesh size |
| IUU fishing | Individual Transferable Quota (ITQ) |
| Coastal development | By-catch reduction device |
| Introduced species | Regional zoning |

**Table 4.** Major reasons for why regulated fisheries are still faced with overexploitation.

|  | **Managers** | **Policy Makers** | **Scientists** | **Fishers** | **NGOs** |
|---|---|---|---|---|---|
| Not enough scientific information | 72% | 54% | 78% | 73% | 80% |
| Scientific knowledge is not fully being used | 64% | 67% | 53% | 62% | 20% |
| Lack of political will | 93% | 92% | 74% | 84% | 80% |

*3.5. Socioeconomic Situations Affecting Fisheries and Marine Systems*

Forty-two percent of the respondents said fish as a protein source is not important for survival in their country, 7% said it was, and 23% considered fish vital for some regions (Appendix A). However, when questioned on how important fishing is as a main source of income, 65% of the respondents said fishing is the major economic activity for a few regions, 42% said fishing is a vital source of income for some regions and 37% said that fishing is somewhat important as a main source of income for the country as a whole (Appendix A). Regarding subsides, 52% of the respondents said that fisheries subsidies are available in their country, 34% said there are no subsidies and 14% did not know (Appendix A). Of those who said there are subsidies in their country, 88% said they have fuel subsidies, 35% have employment subsidies, 26% have lower interest rates on bank loans and 15% said they have subsidies related to culture. Sixty-five percent of the respondents believed that subsidies contribute to overcapacity of the fishing industry (Figure 16).

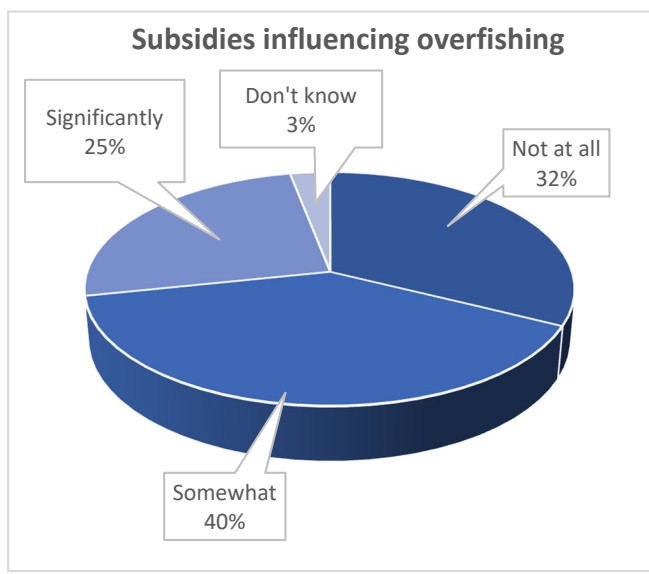

**Figure 16.** Respondents' belief regarding whether subsidies contribute to overcapacity of the fishing industry (n = 87).

There was particular support amongst the respondents for economic incentives, such as fishing access agreements and fishing vessel buy-backs by the government (Figure 17).

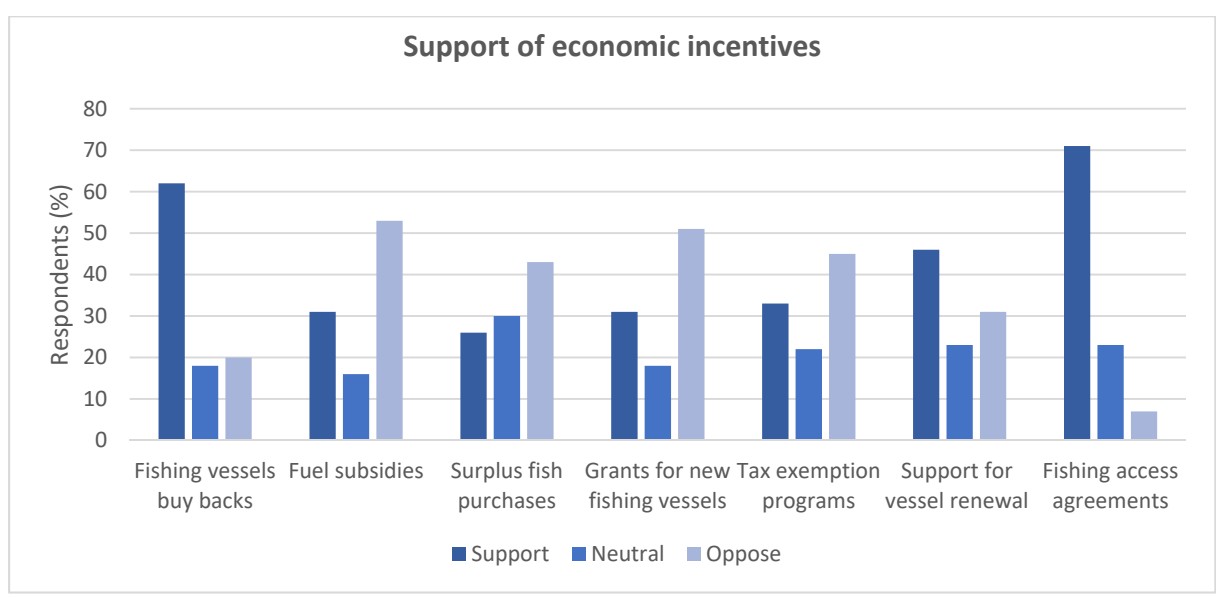

**Figure 17.** Experts showed large support for fishing vessel buy-back schemes and fishing access agreements (n = 168).

Fifty-one percent of the respondents were not able to estimate the cost of management for the fishery they work with (Appendix A).

## 4. Discussion

Results from the survey demonstrate that the respondents have had extensive experience in the fisheries management process, including both science and management. The respondents had formal qualifications and/or experience; with 42% having Doctoral degrees, 28% Masters degrees and almost half of the respondents having senior or executive roles in fisheries. The coverage was also global, representing 34 nations in total. While we acknowledge the sample sizes were uneven, with more scientists answering than any of the other respondents, there was congruence in many results, suggesting that perceptions held by fisheries scientists and managers may not actually be that different. Indeed, in many cases, fishers also held similar attitudes, though there were some notable differences (e.g., on the need for additional enforcement). In following up on why it proves so hard to access the opinions of managers, let alone policy-makers (who were an even smaller respondent group), it became clear that they lack opportunities to gather and share information in the same way as provided by scientific conferences. Funding such travel is often hard to do. In improving the state of fisheries globally—sharing insights into what has and has not worked—it appears that there is a fundamental need for the creation of a fora, or a conduit, for information sharing amongst these managerial and policy groups.

### 4.1. Threats and Challenges in Sustaining Fisheries

This analysis clearly confirmed that sustaining fisheries is a complex challenge, but the experts also offered their opinions as to how to combat the issues involved, which are generally consistent with the literature on how to sustainably manage fisheries [37–40]. The respondents considered the 10 main threats to fisheries to be overfishing, climate change, habitat destruction, pollution, ecosystem shifts, IUU fishing, ocean acidification, costal development, land-based pollution and introduced species. These same threats were considered important at national and global scales. This shows that the threats and challenges to sustaining fisheries are similar around the world; a finding consistent with existing scientific literature [8,41–43].

### 4.2. Management Tools in Sustaining Fisheries

Although the analysis highlights an extensive range of challenges in achieving sustainable fisheries, it also shows that the respondents believe there are many existing tools for addressing these obstacles and supporting sustainable fishing. Just as the main challenges and threats to sustaining fisheries were viewed similarly around the world, so too the list of potential tools was consistent across respondents from differing backgrounds and nationalities. While overfishing was seen as a major threat to sustaining fisheries (nationally and globally), the majority of all responding groups said it is not a challenge to manage. Given concern over the magnitude of the problems facing "small scale" fisheries and the difficulties of achieving successful management in locations with few regulatory resources [44], this is a surprising response. However, this may be because the respondents primarily work in fisheries with a range of regulations in place, with compliance and enforcement mechanisms already implemented to combat this challenge and so they have directly experienced the management of overfishing. This result may highlight a tacit bias in the work—people working in less well-resourced fisheries are unlikely to have had the means to visit the Congress where the survey was undertaken—and future follow-up on this work should endeavour to address this gap.

Tools identified as useful in sustaining fisheries included sound science, input controls (gear restrictions, seasonal closures, spatial closures, spawning closures, by-catch reduction device, size limits and regional zoning), output controls (bag limits, ITQs, Total Catch Limits (TACs)), a mixture of top-down and bottom-up organisation, stakeholder participation, fishing access agreements and fishing vessels buy-backs, effectively taking an integrated or ecosystem approach. In particular, the vast majority of all responding groups viewed good science, MPAs, ITQs, gear restrictions and stakeholder participation to be the five most efficient tools for Ecosystem-Based Fisheries Management. All of these tools are consistent with what have been recorded as good supporting tools for sustainable fisheries in other research [39,45–47].

More of the respondents were satisfied than dissatisfied with the EBFM's planning and implementation processes. More were, however, neutral regarding the results of the EBFM, reflecting in part the complex nature of the EBFM process. Management tools might be put in place, but it may take a long time before any results are seen. These approaches may be introduced when the system has been overfished and shifted to a state where restoration may take a lengthy period [48–50]. More managers than any other responding group said they believed the EBFM implementation process was a success. About the same number of managers, policy-makers and scientists said they believed it was unsuccessful. Possibly, there were different expectations among the various responding groups, where the managers saw it as a success in itself that such a large management process had been adopted and implemented by the government in the first place; while the scientists may have been more cautious (neutral) because any biological success was yet to be seen. More managers and policy-makers said they were satisfied with the results of EBFM than the scientists and fishers, although all responding groups showed a cautious element to any success, the fishers more so than any other group. Again, the expectations are likely to differ among the various stakeholders, as implementing EBFM unavoidably involves trade-offs in meeting all biological, economic and social goals [51], which will differ between the different groups.

Given the growing focus on the implications of a high level of marine pollution [52–54], it might be surprising that only just over half of the respondents answered that they believe land-based pollution is a major threat to the world's fisheries and 46% said plastic is a major threat. This might be due to the fact that the survey was undertaken in 2012 when there was not as much scientific reporting on plastics in the ocean [55]. It was particularly noteworthy though that, despite pollution and plastics being identified as threats, few, if any, of the suggested tools put forward are likely to have a significant role in combating these issues. This indicates that, while awareness of the issue is growing, focus is still on the classical threats and long-established tools.

### 4.3. Management Constraints in Using More Science

Fisheries management in the majority of industrialised nations is said to be science or evidence-based, even if science-based advice is not always followed in the political process [56]. This analysis showed 'not using scientific knowledge to its fullest potential' to be the main constraint for effectively and efficiently implementing ecosystem-based fisheries management, together with: (1) a lack of compliance; (2) IUU still being a major global issue; and (3) political will.

The management of marine systems in general, and fisheries in particular, is highly complex and a story of information paucity. It is very difficult to estimate even the abundance of target species. In some regions, it is even difficult to precisely determine what has been extracted from the ocean, let alone the effects on dependent species or species not directly impacted by fishing [57]. The reason why science is not being used to its fullest is interesting. Is it because of a disconnect of science and management? In Australia, having fisheries scientists work closely with but ultimately sit apart from the management agency has been a successful approach, as the participatory processes in place there allow for communication, while the 'distance' has helped increase trust in science and motivation of scientists by all stakeholders. In other regions, the organisational disconnect has led to barriers to information uptake. In these latter instances, because scientists belong to a separate organisation, they are treated more as a consultant and thereby not fully integrated in the management process, leading to critical communication failures. An example of this is where scientists from the International Council for the Exploration of the Sea (ICES) advise the Oslo Paris Commission (OSPAR), the Helsinki Commission, the Baltic Marine Environment Protection Commission (HELCOM), the North East Atlantic Fisheries Commission (NEAFC), the North Atlantic Salmon Conservation Organization (NASCO) and the European Commission (EC) [58]. Yet, despite all of these channels, the decisions have still been largely political, leading to overfishing within the European Union [59–62]. More recently, there have been significant efforts to reverse this, though it has only been patchily effective; the Mediterranean, in particular, still has a majority of its stocks in an overfished state [63].

An alternative example is found with the Commission for the Conservation for Antarctic Marine Living Resources (CCAMLR). CCAMLR has its scientific committee with its working groups fully integrated in the organisation advising the commission at the annual meetings. Many participants are a part of both the scientific commission and the commission [64–67]. This science-based commitment to ecosystem-based management has, since 1982 (when CCAMLR was founded), contributed to the recovery of previous overfished stocks, and sustainable management of the Southern ocean ecosystems, including fisheries [39,68,69].

### 4.4. A Brief Comment on Cognitive Inconsistencies

With the growing accessibility of literature regarding human cognition, it would be remiss of us not to note how the perceptions reported in this survey may be effected by common cognitive biases and fallacies [70,71]. We are not trained professionals in the field of psychology, so will not go into depth, but the results for IUU appear to be a stand out example of such biases in action. There is clear recognition that IUU is a problem, with almost complete consensus on this point across respondents. However, it appears that the perception of the magnitude of the problem is strongly influenced by an optimism bias (with far fewer respondents thinking it is a problem in their own fishery) and by biases to do with framing (it is seen as more of an issue when asked directly about IUU rather than in general bundled with other risks) and uncertainty (as the true magnitude of the problem is typically unknown and so may be discounted as a result). In addition, the fact that the suggested solutions for sustainable fisheries include a list of existing tools, many of which have been in use in fisheries for centuries, suggest that there may be a strong endowment effect, with experts sticking strongly to tools they are already heavily invested in without necessarily looking for new alternatives. This is worth additional research to verify. If confirmed, it would open up new research paths; if falsified, then it would reassure all stakeholders that we already have at hand all the tools we need to achieve sustainable fisheries.

*4.5. Political Will to Match Biological Challenges*

The survey showed that, despite implementation of EBFM and increased levels of input from science, industry and NGOs, sustaining fisheries remains a challenge. The main challenge when managing fisheries was said to be a lack of political will. We note that policy-makers represented just 7% of the respondents, and the issue of sustaining fisheries due to a lack of political will might have been viewed differently had there been more policy people participating in the survey. Indeed, knowledge brokers who span the science–policy interface caution that policy-makers can become frustrated with scientists who fail to appreciate the many sources of information and many pressures that must be navigated by policy-makers when making a single decision [72]. Political advisers and politicians must also consider political, social, cultural and economic matters.

The challenge to managing fisheries ranked second by the respondents was a shortage in compliance and regulations, stock assessments and monitoring. This might not come as a surprise as there are high costs involved for scientific assessments and controlling regulations [73]. In linking the top two challenges, the challenge found regarding the lack of compliance may reflect a lack of general political and social will to fund and implement required management controls [70]. Politicians may be more inclined to act on issues more important to the voters (who have concerns extending well beyond fisheries), and perhaps, at times, they do not either fully appreciate the seriousness of the marine issues or the need for long-term sustainable plans that span many election cycles.

However, what might not be high on the political agenda today may change with building public awareness, which in turn may demand better management of natural resources [71]. The United Nations' Ocean Conference for implementation of Sustainable Development Goal 14 ('Conserve and sustainably use the oceans, seas and marine resources for the sustainable development') is an example. This conference was held in June 2017, with 193 nations making a commitment to a set of measures aiming to increase the resilience of ocean health. These pledges have been accompanied by over 1400 voluntary commitments. Together, these commitments can be seen as a global commitment (raised from increased scientific and public pressure) for politicians to better manage marine life. Given increased consciousness of environmental issues among the public since this survey was conducted [72,73], it would be interesting to conduct a similar survey today to see if there is a perception of a stronger political will today to sustain fisheries.

## 5. Conclusions

This study reinforces the magnitude of the challenges in sustaining fisheries. It identified key issues underpinning the use of an ecosystem management approach, such as complexity, the high degree of connectivity, difficulties associated with observing ocean processes and monitoring flora and fauna. The fact that 99% of the respondents believed that IUU fishing still is a global problem and 65% estimated the global level of IUU fishing to be between 31 and 60% of the total catch worldwide is, naturally, a major concern. Tools identified as useful in sustaining fisheries included sound science, gear restrictions, seasonal closures, spatial closures, spawning closures, by-catch reduction device, size limits and regional zoning, bag limits, ITQs and TACs. The study indicated that the common position of the respondents is that the use of a mixture of top-down and bottom-up organisation and institutional forms is important to success, as is the importance of stakeholder participation. However, implementing these solutions will come with new challenges, especially when implementing them at scales aligning with the magnitude of participation in "small-scale" (often poorly resourced) fisheries in developing nations. The survey also highlighted the impact of fishing access agreements and fishing vessels buy-backs as tools to constrain effort. Again, these are things that may work more effectively for industrial than some artisanal fisheries.

This research illustrated a clear perception of a need for a higher political will and commitment to combat challenges, such as IUU fishing, habitat destruction and climate change, both nationally and globally. More research and long-term monitoring to assist managers in prioritization resources was also identified as a particularly important need. It was clear from the analysis that the widely

held belief by those experts in charge of the world's fisheries that, to recover from overfishing and fisheries collapse (and to minimise the future risk of such events), scientific input must be matched with the same level of political commitment, including implementing science-based fisheries and conservation measures.

It is also worth noting that human cognition is not infallible. When asked directly about illegal, unreported and unregulated fishing, 99% of the respondents saw it as a global issue; however, when put against other challenges, close to 70% of the policy-makers and scientists believed that is not a major threat to national fisheries, despite the fact that almost 80% of the fishers said they think it is. This suggests that there is a gap in the discourse and management of IUU fishing that likely needs closer consideration or discussion.

This analysis showed that there is the strong perception that scientific knowledge is not being used to its fullest potential and that in turn is the main constraint for effectively and efficiently implementing ecosystem-based fisheries management. Is the challenge then a lack of political will only, or is this a reflection of the make-up of respondents: scientists frustrated with a perceived lack of political appreciation? Perhaps there is a greater need to establish science-management networks that meet regularly, to train a new generation of scientists who have direct industry and regulatory body experience (spending time in both as well as academia before completing their training), as well as a need for scientists to communicate science in a more pedagogical way?

**Author Contributions:** Conceptualization, J.A.N. and E.A.F.; Methodology, J.A.N. and E.A.F.; Software, J.A.N.; Validation, J.A.N. and C.R.J.; Formal Analysis, J.A.N., E.A.F. and C.R.J.; Investigation, J.A.N.; Resources, J.A.N. and E.A.F.; Data Curation, J.A.N. and E.A.F.; Writing-Original Draft Preparation, J.A.N.; Writing-Review & Editing, J.A.N. and E.A.F., C.R.J. and M.H.; Visualization, J.A.N.; Supervision, J.A.N. and E.A.F., C.R.J. and M.H.; Project Administration, J.A.N.

**Funding:** This research received no external funding.

**Conflicts of Interest:** The authors declare no conflict of interest.

**Appendix A**

Fisheries Governance Survey, with responses

**Q1.** Threats to the marine environment: For each of the potential marine threats, please tell if you believe there is no threat, a minor threat or a major threat.

| Responses to the Fisheries Governance Survey are Presented in the Order the Questions Appeared in the Survey Instrument. I Have Read the Information Above and Consent to Participate in This Study. I am over the Age of 18 Years. Answer | Response | % |
|---|---|---|
| Yes | 188 | 100 |
| No | 0 | 0 |
| Total | 188 | 100 |

No threat

| Question | National Fisheries | World Fisheries | Total Responses |
|---|---|---|---|
| Pollution sourced from land | 9 | 4 | 13 |
| Eutrophication | 19 | 16 | 35 |
| Anoxic events | 23 | 20 | 43 |
| Ocean acidification | 14 | 8 | 22 |
| Introduced species and pests | 5 | 5 | 10 |
| Dead marine zones | 25 | 14 | 39 |
| Energy exploration | 33 | 21 | 54 |
| Ecosystem shifts | 11 | 5 | 16 |
| Habitat destruction | 8 | 0 | 8 |

| Question | National Fisheries | World Fisheries | Total Responses |
|---|---|---|---|
| Plastics in the oceans | 23 | 12 | 35 |
| Coastal development | 14 | 16 | 30 |
| Overfishing | 12 | 0 | 12 |
| Climate change | 6 | 3 | 9 |
| IUU fishing | 9 | 1 | 10 |

Minor threat

| Question | National Fisheries | World Fisheries | Total Responses |
|---|---|---|---|
| Pollution sourced from land | 83 | 65 | 148 |
| Eutrophication | 95 | 76 | 171 |
| Anoxic events | 95 | 78 | 173 |
| Ocean acidification | 79 | 61 | 140 |
| Introduced species and pests | 91 | 79 | 170 |
| Dead marine zones | 92 | 83 | 175 |
| Energy exploration (oil, gas, etc.) | 87 | 84 | 171 |
| Ecosystem shifts | 74 | 63 | 137 |
| Habitat destruction | 57 | 41 | 98 |
| Plastics in the oceans | 94 | 62 | 156 |
| Coastal development | 75 | 61 | 136 |
| Overfishing | 49 | 32 | 81 |
| Climate change | 63 | 46 | 109 |
| IUU fishing | 40 | 14 | 54 |
| Other, please specify | 4 | 5 | 8 |

Major threat

| Question | National Fisheries | World Fisheries | Total Responses |
|---|---|---|---|
| Pollution sourced from land | 78 | 98 | 176 |
| Eutrophication | 56 | 65 | 121 |
| Anoxic events | 48 | 53 | 101 |
| Ocean acidification | 65 | 96 | 161 |
| Introduced species and pests | 72 | 78 | 150 |
| Dead marine zones | 46 | 63 | 109 |
| Energy exploration (oil, gas, etc.) | 45 | 63 | 108 |
| Ecosystem shifts | 78 | 97 | 175 |
| Habitat destruction | 98 | 123 | 221 |
| Plastics in the oceans | 49 | 87 | 136 |
| Coastal development | 76 | 85 | 161 |
| Overfishing | 103 | 141 | 244 |
| Climate change | 95 | 119 | 214 |
| IUU fishing | 47 | 86 | 133 |
| Other, please specify | 13 | 19 | 32 |

**Q2.** In your experience, what are the three main challenges of managing fisheries? Please add a brief description.

| Answer | Response | % |
|---|---|---|
| Lack of political will | 98 | 56% |
| Not all stake holders are involved | 34 | 20% |
| Not enough compliance with regulations | 57 | 33% |
| Fisheries are very complex to manage | 29 | 17% |
| International cooperation is needed | 25 | 14% |
| Over-fishing | 51 | 29% |

| Answer | Response | % |
|---|---|---|
| Lack of knowledge in fish behaviour | 11 | 6% |
| High amounts of by-catch and discard | 30 | 17% |
| Poverty | 14 | 8% |
| Stock assessment and monitoring | 49 | 28% |
| Need to track trading of fish products | 12 | 7% |
| Growing human population (food security) | 22 | 13% |
| Take high levels of uncertainty into account when setting quotas | 12 | 7% |
| Ecosystem management | 24 | 14% |
| Consider socio-economic implications in poorer regions | 21 | 12% |
| Impacts of climate change | 20 | 11% |
| Amount of IUU fishing is underestimated | 37 | 21% |
| Stakeholder agreements | 19 | 11% |
| Other | 39 | 22% |

**Q3.** In what country do you work?

| Answer | Response | % |
|---|---|---|
| Argentina | 2 | 1% |
| Australia | 40 | 24% |
| Bangladesh | 1 | 1% |
| Canada | 5 | 3% |
| China | 1 | 1% |
| Czech Republic | 1 | 1% |
| Denmark | 1 | 1% |
| France | 4 | 2% |
| Germany | 2 | 1% |
| Greece | 1 | 1% |
| Iceland | 4 | 2% |
| India | 1 | 1% |
| Indonesia | 2 | 1% |
| Ireland | 1 | 1% |
| Italy | 3 | 2% |
| Japan | 3 | 2% |
| Kenya | 1 | 1% |
| Mexico | 3 | 2% |
| Mongolia | 1 | 1% |
| Namibia | 5 | 3% |
| Netherlands | 3 | 2% |
| New Zealand | 2 | 1% |
| Nigeria | 5 | 3% |
| Norway | 2 | 1% |
| Philippines | 2 | 1% |
| Saudi Arabia | 1 | 1% |
| South Africa | 5 | 3% |
| Spain | 1 | 1% |
| Sweden | 8 | 5% |
| Tanzania | 1 | 1% |
| Turkey | 2 | 1% |
| Uganda | 1 | 1% |
| United Kingdom | 30 | 18% |
| United States | 21 | 12% |
| Total | 170 | 100% |

**Q4.** What is your role in fisheries?

| Answer | Response | % |
|---|---|---|
| Fisheries manager/Natural resource manager | 14 | 8% |
| Fisher | 31 | 18% |
| Policy maker | 13 | 7% |
| Scientist | 96 | 54% |
| NGO member | 5 | 3% |
| Other, please specify | 18 | 10% |
| Total | 177 | 100% |

**Q5.** Where do you work?

| Answer | Response | % |
|---|---|---|
| National management | 40 | 34% |
| Sub-national management | 15 | 13% |
| Community/Communal/Indigenous | 2 | 2% |
| International | 28 | 24% |
| University | 17 | 15% |
| Other, please specify | 15 | 13% |
| Total | 117 | 100% |

**Q6.** What position/level do you work at now?

| Answer | Response | % |
|---|---|---|
| Field management | 28 | 19% |
| Middle management | 50 | 34% |
| Senior management | 51 | 35% |
| Executive management | 17 | 12% |
| Total | 146 | 100% |

**Q7.** What fishery or fisheries are you involved in? If you work with several fisheries, please pick one fishery. Should you wish to give information about more than one fishery, please take the survey again?

| Answer | Response | % |
|---|---|---|
| Large pelagic | 23 | 16% |
| Small pelagic | 22 | 15% |
| Large demersal | 36 | 25% |
| Small demersal | 10 | 7% |
| Crustaceans | 17 | 12% |
| Shellfish | 2 | 1% |
| Inland fishery | 3 | 2% |
| Aquaculture | 4 | 3% |
| Coastal | 12 | 8% |
| Shark | 1 | 1% |
| Other | 13 | 9% |
| Total | 143 | 100% |

**Q8.** How would you best describe the fishery you work in?

| Answer | Response | % |
|---|---|---|
| Collapsed | 10 | 6% |
| Highly overfished | 15 | 9% |
| Overfished | 49 | 28% |
| Sustainably fished | 67 | 39% |
| Recovering | 14 | 8% |

| Answer | Response | % |
|---|---|---|
| Developing/exploratory | 4 | 2% |
| No information | 13 | 8% |
| Total | 172 | 100% |

**Q9.** How many years of experience do you have in fisheries?

| Answer | % |
|---|---|
| 0–3 years | 16% |
| 3–5 years | 10% |
| 5–10 years | 11% |
| 10–15 years | 14% |
| 15–20 years | 17% |
| 20–25 years | 17% |
| More than 25 years | 15% |

**Q10.** What are the major changes that have occurred in fisheries management during your career with fisheries? Multiple answers possible.

| Answer | Response | % |
|---|---|---|
| There are no major changes | 8 | 7% |
| Increased level of scientific input | 60 | 55% |
| Increased level of industry input | 53 | 49% |
| Increased level of NGO input | 47 | 43% |
| Environmental versus fisheries department | 40 | 37% |
| Level of collaboration amongst stake holders and organizations | 51 | 47% |
| Increased number of staff | 8 | 7% |
| Increased number of scientists | 26 | 24% |
| Amount of resources (money, staff) | 18 | 17% |
| Ecosystem based management instead of single species management | 50 | 46% |
| Dealing with pollution (e.g., terrestrial run-offs like fertilizer, soil turbidity) | 16 | 15% |
| Other, please specify | 20 | 19% |

**Q11.** In the last 5–10 years, have resources (such as funding, staff, research, equipment) for management overall:

| Answer | Response | % |
|---|---|---|
| Increased a lot | 5 | 4% |
| Increased a little | 49 | 39% |
| Stayed about the same | 35 | 28% |
| Decreased a little | 25 | 20% |
| Decreased a lot | 12 | 10% |
| Total | 126 | 100% |

**Q12.** Has the fishery you work with implemented Ecosystem-Based Fisheries Management (EBFM) or a similar holistic approach to governing fisheries?

| Answer | Response | % |
|---|---|---|
| Yes | 104 | 60% |
| No | 68 | 40% |
| Total | 172 | 100% |

**Q13.** How well do you consider the overall implementation process of EBFM, or similar management approach, to have gone?

| Answer | Response | % |
|---|---|---|
| Very successful | 11 | 10% |
| Successful | 32 | 30% |

| Answer | Response | % |
|---|---|---|
| Neutral | 50 | 47% |
| Unsuccessful | 13 | 12% |
| Very unsuccessful | 1 | 1% |
| Total | 107 | 100% |

**Q14.** How satisfied are you with the Ecosystem-Based Fisheries Management process?

| Question | Very Satisfied | Satisfied | Neutral | Dissatisfied | Very Dissatisfied | Total Responses |
|---|---|---|---|---|---|---|
| Planning process | 11 | 47 | 33 | 9 | 4 | 104 |
| Implementation process | 8 | 40 | 30 | 23 | 3 | 104 |
| Results | 7 | 26 | 45 | 21 | 4 | 103 |

**Q15.** Briefly describe your experience with the implementation of EBFM.

| Answer | Response | % |
|---|---|---|
| It still doesn't consider the whole ecosystem | 30 | 36% |
| Lack in scientific knowledge delays proper implementation | 19 | 23% |
| Highly complex procedure, which makes it hard to really implement EBFM | 48 | 58% |
| Lack of compliance to secure successful EBFM | 22 | 27% |
| Time consuming | 19 | 23% |
| Difficult to decide what variables and what species (spp). Species should be considered as there are so many variables and spp in an ecosystem | 28 | 34% |
| Insufficient compliance | 10 | 12% |
| It has worked very well | 6 | 7% |
| Improvements can already be seen | 15 | 18% |
| It has been a satisfactory process | 11 | 13% |
| Other | 11 | 13% |

**Q16.** How do you view the role of governance and management to fisheries in your country as well as worldwide? For each of the following variables, please say if you believe there is a need for more or less of the following variables.

Highly needed

| Variables | National Fisheries | World Fisheries | Total Responses |
|---|---|---|---|
| Stronger political will to manage fisheries | 98 | 131 | 229 |
| Improved conservation measures | 68 | 107 | 175 |
| Enforcement of regulations | 69 | 112 | 181 |
| Change of governance structure | 57 | 86 | 143 |
| More money | 59 | 81 | 140 |
| More staff | 51 | 74 | 125 |
| More research | 71 | 98 | 169 |
| More international collaboration | 83 | 116 | 199 |
| Managing Illegal, Unreported and Unregulated fishing (IUU) | 76 | 128 | 204 |

Somewhat needed

| Variables | National Fisheries | World Fisheries | Total Responses |
|---|---|---|---|
| Stronger political will to manage fisheries | 36 | 27 | 63 |
| Improved conservation measures | 56 | 46 | 102 |
| Enforcement of regulations | 50 | 39 | 89 |
| Change of governance structure | 58 | 55 | 113 |
| More money | 76 | 63 | 139 |
| More staff | 70 | 59 | 129 |

| Variables | National Fisheries | World Fisheries | Total Responses |
|---|---|---|---|
| More research | 65 | 51 | 116 |
| More international collaboration | 48 | 31 | 79 |
| Managing Illegal, Unreported and Unregulated fishing (IUU) | 47 | 30 | 77 |

Satisfactory as it is

| Variables | National Fisheries | World Fisheries | Total Responses |
|---|---|---|---|
| Stronger political will to manage fisheries | 19 | 5 | 24 |
| Improved conservation measures | 28 | 4 | 32 |
| Enforcement of regulations | 38 | 9 | 47 |
| Change of governance structure | 35 | 11 | 46 |
| More money | 28 | 10 | 38 |
| More staff | 38 | 19 | 57 |
| More research | 23 | 8 | 31 |
| More international collaboration | 21 | 9 | 30 |
| Managing Illegal, Unreported and Unregulated fishing (IUU) | 32 | 3 | 35 |

Less needed

| Variables | National Fisheries | World Fisheries | Total Responses |
|---|---|---|---|
| Stronger political will to manage fisheries | 8 | 3 | 11 |
| Improved conservation measures | 10 | 3 | 13 |
| Enforcement of regulations | 3 | 1 | 4 |
| Change of governance structure | 8 | 1 | 9 |
| More money | 3 | 2 | 5 |
| More staff | 6 | 2 | 8 |
| More research | 3 | 0 | 3 |
| More international collaboration | 7 | 2 | 9 |
| Managing Illegal, Unreported and Unregulated fishing (IUU) | 1 | 0 | 1 |

**Q17.** Why do you believe, on a global scale, we are still facing fisheries overexploitation in regulated fisheries? Drag and drop your rankings.

| Question | Major Challenge | Some Challenge | Minor Challenge | No Challenge | Total Responses |
|---|---|---|---|---|---|
| There is not enough scientific information. | 43 | 74 | 40 | 4 | 161 |
| Scientific knowledge is not being used to its fullest. | 90 | 49 | 21 | 2 | 162 |
| Lack of political will. | 133 | 25 | 10 | 0 | 168 |
| There needs to be stricter laws and regulations. | 74 | 63 | 24 | 4 | 165 |
| There needs to be more compliance and enforcement of laws. | 109 | 45 | 11 | 1 | 166 |
| Management is focused on species rather than eco-based management. | 81 | 58 | 20 | 5 | 164 |
| General public does not care enough about sustainable fishing to make it worthwhile for politicians to make it a priority. | 68 | 60 | 31 | 7 | 166 |
| Fish abundance is too complex to predict. | 39 | 70 | 50 | 7 | 166 |
| Lack of formal harvest strategies | 44 | 66 | 45 | 7 | 162 |
| Environmental variables affecting fisheries abundance are too complex to measure and predict. | 50 | 66 | 39 | 9 | 164 |
| Commercial fishers have too much influence. | 54 | 62 | 31 | 16 | 163 |
| There is not enough scientific expertise to interpret scientific data on management level. | 47 | 54 | 50 | 13 | 164 |
| Lack of political knowledge on marine and fisheries related issues. | 87 | 55 | 17 | 3 | 162 |
| Other | 18 | 2 | 0 | 0 | 20 |

**Q18.** What management tools are being and should be used to manage the fishery you work in?

| Question | Tools Being Used | Tools That Should Be Used | Total Responses |
|---|---|---|---|
| Total Allowable Catch (TAC) | 116 | 53 | 169 |
| Individual Transferable Quota (ITQ) | 66 | 47 | 113 |
| Seasonal closures | 104 | 68 | 172 |
| Regional zoning | 66 | 46 | 112 |
| Spatial closures (e.g., MPA) | 95 | 63 | 158 |
| Spawning closures | 69 | 60 | 129 |
| Size limits | 99 | 70 | 169 |
| Commercial only fishing areas | 19 | 23 | 42 |
| Recreation only fishing areas | 23 | 28 | 51 |
| Ecosystem based management | 67 | 73 | 140 |
| Bag limits | 38 | 36 | 74 |
| Mesh size | 75 | 53 | 128 |
| Trawling net size restrictions | 59 | 34 | 93 |
| Fishing vessel size restriction | 38 | 25 | 63 |
| Horsepower restrictions | 26 | 20 | 46 |
| Tabu/Taboo | 9 | 9 | 18 |
| Bottom trawling is banned | 34 | 33 | 67 |
| Other gear restrictions | 65 | 29 | 94 |
| Fishing vessels buy backs by government | 16 | 15 | 31 |
| Fuel subsidies | 35 | 18 | 53 |
| Surplus fish purchases | 11 | 22 | 33 |
| Grants for new fishing vessels | 18 | 12 | 30 |
| Tax exemption programs | 13 | 14 | 27 |
| Vessel construction, renewal and modernization | 20 | 15 | 35 |
| Fishing access agreements | 25 | 23 | 48 |
| By-catch reduction device | 59 | 46 | 105 |
| Other | 9 | 13 | 22 |

**Q19.** In your work, who is and who should be involved in the fisheries management process?

| Question | Who is Involved? | Who Should be Involved? | Total Responses |
|---|---|---|---|
| Fisheries managers | 148 | 86 | 234 |
| Natural resource managers | 75 | 80 | 155 |
| Fishers | 103 | 103 | 206 |
| Politicians | 130 | 67 | 197 |
| Scientists | 133 | 95 | 228 |
| NGOs | 80 | 78 | 158 |
| The public | 35 | 69 | 104 |
| Local communities | 36 | 79 | 115 |
| Other | 3 | 6 | 9 |

**Q20.** Here is a range of input controls used in fisheries management. Do you support/oppose the concept of?

| Question | Strongly Support | Support | Neutral | Oppose | Strongly Oppose | Total Responses |
|---|---|---|---|---|---|---|
| Gear restrictions | 105 | 43 | 16 | 1 | 1 | 166 |
| Vessel size restrictions | 51 | 40 | 38 | 30 | 4 | 163 |
| Horsepower restrictions | 38 | 35 | 50 | 35 | 5 | 163 |
| Seasonal closures | 107 | 45 | 12 | 2 | 0 | 166 |
| Regional zoning | 87 | 47 | 25 | 3 | 0 | 162 |
| Recreational only fishing areas | 42 | 33 | 56 | 24 | 6 | 161 |
| Spatial closures | 105 | 47 | 12 | 1 | 0 | 165 |

| Question | Strongly Support | Support | Neutral | Oppose | Strongly Oppose | Total Responses |
|---|---|---|---|---|---|---|
| Spawning closures | 109 | 37 | 14 | 1 | 0 | 161 |
| Size limits | 100 | 42 | 20 | 2 | 1 | 165 |
| Commercial only fishing areas | 38 | 36 | 58 | 28 | 0 | 160 |
| BRDs (by-catch reduction device) | 100 | 48 | 12 | 2 | 0 | 162 |

**Q21.** There is a range of output controls used in fisheries management. Do you support/oppose the concept of?

| Question | Strongly Support | Support | Neutral | Oppose | Strongly Oppose | Total Responses |
|---|---|---|---|---|---|---|
| Total Catch Limits (TACs) | 100 | 43 | 22 | 2 | 1 | 168 |
| Individual Transferable Quotas (ITQ) | 75 | 41 | 40 | 7 | 5 | 168 |
| Bag limits | 71 | 44 | 45 | 4 | 1 | 165 |

**Q22.** In your experience in fisheries, do you support/oppose the concept of?

| Question | Strongly Support | Support | Neutral | Oppose | Strongly Oppose | Total Responses |
|---|---|---|---|---|---|---|
| Fishing vessels buy backs by government | 40 | 64 | 30 | 25 | 9 | 168 |
| Fuel subsidies | 33 | 19 | 26 | 36 | 52 | 166 |
| Surplus fish purchases | 13 | 30 | 50 | 38 | 34 | 165 |
| Grants for new fishing vessels | 31 | 21 | 30 | 35 | 50 | 167 |
| Tax exemption programs | 29 | 26 | 36 | 31 | 44 | 166 |
| Vessel construction, renewal and modernization | 34 | 43 | 39 | 16 | 35 | 167 |
| Fishing access agreements | 57 | 61 | 38 | 7 | 4 | 167 |

**Q23.** How much do you estimate the fishery you work with costs to manage annually (US dollar)? Costs include research, management, subsidies.

| Answer | Response | % |
|---|---|---|
| <US$500,000 | 11 | 7% |
| US$500,000–1 million | 18 | 11% |
| US$1–$2 million | 6 | 4% |
| US$3–5 million | 16 | 10% |
| US$6–15 million | 6 | 4% |
| US$16–20 million | 6 | 4% |
| US$21–30 million | 1 | 1% |
| US$31–40 million | 1 | 1% |
| US$41–50 million | 1 | 1% |
| US$51–60 million | 2 | 1% |
| US$61–70 million | 1 | 1% |

| Answer | Response | % |
|---|---|---|
| US$71–80 million | 0 | 0% |
| US$81–90 million | 2 | 1% |
| US$91–100 million | 2 | 1% |
| US$101–150 million | 1 | 1% |
| US$151–200 million | 2 | 1% |
| US$200–250 million | 1 | 1% |
| >US$ 250 million | 4 | 2% |
| Local currency, if you wish | 0 | 0% |
| Don't know | 86 | 51% |
| Total | 167 | 100% |

**Q24.** Do you know how much revenue your fishery provide annually?

| Answer | Response | % |
|---|---|---|
| Yes | 39 | 31% |
| No | 87 | 69% |
| Total | 126 | 100% |

**Q25.** How many fishing vessels operate within your fishery?

| Answer | Response | % |
|---|---|---|
| 1–5 | 19 | 13% |
| 6–25 | 33 | 23% |
| 26–50 | 22 | 15% |
| 51–75 | 13 | 9% |
| 76–100 | 5 | 4% |
| >100 | 50 | 35% |
| Total | 142 | 100% |

**Q26.** How many fishing vessels are registered in the country where you work?

| Answer | Response | % |
|---|---|---|
| 1–10 | 5 | 9% |
| 11–30 | 1 | 2% |
| 31–60 | 2 | 4% |
| 61–100 | 2 | 4% |
| 101–200 | 3 | 5% |
| 201–400 | 3 | 5% |
| 401–600 | 6 | 11% |
| 601–1000 | 2 | 4% |
| 1001–2000 | 8 | 14% |
| 2001–5000 | 9 | 16% |
| 5001–10,000 | 5 | 9% |
| 10,001–20,000 | 7 | 13% |
| >20,000 | 3 | 5% |
| Total | 56 | 100% |

**Q27.** In your country, how important is fishing as a main food source of protein?

| Answer | Response | % |
|---|---|---|
| Overall survival depends on fishing | 12 | 7% |
| Vital for some regions/areas | 39 | 23% |
| Somewhat important | 46 | 27% |
| Not important for survival | 71 | 42% |
| Total | 168 | 100% |

**Q28.** In your country, how important is fishing as a main source of income?

| Answer | Response | % |
|---|---|---|
| Overall income depends on fishing | 8 | 5% |
| Vital for some regions/areas | 70 | 42% |
| Somewhat important | 61 | 37% |
| Not important for income | 27 | 16% |
| Total | 166 | 100% |

**Q29.** In your country, are there regions where fishing is the major economic activity?

| Answer | Response | % |
|---|---|---|
| Yes, many regions | 29 | 18% |
| Yes, a few regions | 107 | 65% |
| Yes, one region | 5 | 3% |
| No | 24 | 15% |
| Total | 165 | 100% |

**Q30.** In your country, are there regions or areas where fishing is the major food source of protein?

| Answer | Response | % |
|---|---|---|
| Yes | 68 | 41% |
| No | 96 | 59% |
| Total | 164 | 100% |

**Q31.** Are subsidies provided for fishers in the country in which you work (including fuel rebates, low interest loans, employment, buy-backs, reduced tax)?

| Answer | Response | % |
|---|---|---|
| Yes | 87 | 52% |
| No | 56 | 34% |
| Don't know | 23 | 14% |
| Total | 166 | 100% |

**Q32.** What type of subsidies are there?

| Answer | Response | % |
|---|---|---|
| Fuel | 75 | 88% |
| Lower interest on bank loans | 22 | 26% |
| Employment payments from the government | 30 | 35% |
| Cultural subsidies | 13 | 15% |
| Other, please specify | 22 | 25% |

**Q33.** Do you believe these subsidies contribute to overcapacity of the fishing industry?

| Answer | Response | % |
|---|---|---|
| Not at all | 28 | 32% |
| Somewhat | 34 | 39% |
| Significantly | 22 | 25% |
| Don't know | 3 | 3% |
| Total | 87 | 100% |

**Q34.** Who should carry the real cost of fish products? Costs include governance, management, research and monitoring of fisheries.

| Answer | Response | % |
|---|---|---|
| Fishers | 113 | 69% |
| Consumers | 112 | 69% |
| Government | 104 | 64% |
| Don't know | 14 | 9% |

**Q35.** The fishery I work with has:

| Answer | Response | % |
|---|---|---|
| A single species management approach | 57 | 37% |
| An ecosystem management approach | 87 | 56% |
| Don't know | 12 | 8% |
| Total | 156 | 100% |

**Q36.** In your experience with fisheries, which five (if any) fisheries management and governance regulations are the most efficient for Ecosystem-Based Fisheries Management?

| Answer | Response | % |
|---|---|---|
| Food and Agriculture Organization of the United Nations code of conduct | 7 | 6% |
| MPAs | 63 | 52% |
| ITQs | 59 | 49% |
| Gear restrictions | 56 | 46% |
| Stakeholder participation | 43 | 36% |
| Good science | 64 | 53% |
| Co-management | 30 | 25% |
| Closures | 28 | 23% |
| No bottom trawling | 25 | 21% |
| Stakeholders' education | 23 | 19% |
| Size limits | 10 | 8% |
| More legislation | 8 | 7% |
| Assessment of implementations | 25 | 21% |
| Spawning closures | 11 | 9% |
| Mesh size | 11 | 9% |
| TAC | 31 | 26% |
| Monitoring | 30 | 25% |
| By-catch Reduction Device (BRD) | 35 | 29% |
| Other | 20 | 17% |

**Q37.** What type of organisation do you believe would be optimal to ensure successful Ecosystem-Based Fisheries Management (or the alike management)?

| Answer | Response | % |
|---|---|---|
| Top-down management (centralised governance) | 11 | 7% |
| Bottom-up management (communal, local) | 13 | 8% |
| Mix of top-down and bottom-up management | 132 | 83% |
| Don't know | 7 | 4% |

**Q38.** Decision making process; information and decisions. For the following statements, please indicate if you agree or disagree.

| Question | Strongly Agree | Agree | Neutral | Disagree | Strongly Disagree | Total Responses |
|---|---|---|---|---|---|---|
| In your role, the scientific information is easy to understand, interpret and apply. | 23 | 74 | 18 | 47 | 2 | 164 |
| You have an appropriate amount of information (scientific or otherwise) to make sound fisheries management decisions. | 27 | 63 | 38 | 31 | 4 | 163 |

| Question | Strongly Agree | Agree | Neutral | Disagree | Strongly Disagree | Total Responses |
|---|---|---|---|---|---|---|
| You consider there are robust mechanisms to deal with assessing uncertainty. | 13 | 64 | 29 | 56 | 2 | 164 |
| You believe you can influence final fisheries management decisions. | 15 | 65 | 27 | 40 | 16 | 163 |
| You believe the current decision making process of your fishery is adequate for sustainable fisheries. | 10 | 57 | 28 | 50 | 17 | 162 |
| Do you believe the current decision making process of your fishery is adequate for an overall sustainable marine biodiversity? | 10 | 45 | 34 | 56 | 17 | 162 |
| Comment | 1 | 1 | 0 | 2 | 1 | 5 |

**Q39.** What information or decision-making processes would you like to see more of when making fisheries or ecosystem management decision?

| Answer | Response | % |
|---|---|---|
| Use of indicators in decision-making process | 31 | 21% |
| More research about ecosystem processes and functions | 41 | 28% |
| Politicians need to understand the science | 62 | 42% |
| All stake-holder involvement | 56 | 38% |
| Industry compliance of regulations | 23 | 16% |
| Supporting fishers with knowledge and implementation of regulations | 23 | 16% |
| Holistic objectives; marine and socioeconomic issues | 34 | 23% |
| Use of EBFM models | 29 | 20% |
| Decreasing IUU fishing | 28 | 19% |
| Integrating fishing and environmental policies | 44 | 30% |
| Political commitment | 52 | 36% |
| Management transparency | 56 | 38% |
| Other | 13 | 9% |

**Q40.** What variables are considered and should be considered when setting fisheries quotas?

| Question | Variables That Are Considered | Variables That Should Be Considered | Total Responses |
|---|---|---|---|
| Size structure of the stock | 117 | 81 | 198 |
| Age structure of the stock | 101 | 81 | 182 |
| Catch data | 122 | 73 | 195 |
| Catch Per Unit Effort (CPUE) | 106 | 67 | 173 |
| Life history traits | 60 | 86 | 146 |
| Maximum Sustainable Yield | 80 | 68 | 148 |
| Maximum Economic Yield | 37 | 52 | 89 |
| Climate change | 23 | 101 | 124 |
| Recruitment | 90 | 92 | 182 |

| Question | Variables That Are Considered | Variables That Should Be Considered | Total Responses |
|---|---|---|---|
| Abundance | 104 | 71 | 175 |
| Mortality | 94 | 73 | 167 |
| Effects on the ecosystem | 41 | 103 | 144 |
| Other, please specify | 7 | 16 | 23 |
| Other, please specify | 2 | 4 | 6 |
| Other, please specify | 2 | 2 | 4 |
| Don't know | 5 | 3 | 8 |

**Q41.** If any, what resources would you like to have more of in order to improve sustainable fisheries and marine biodiversity?

| Answer | Response | % |
|---|---|---|
| Resources are already adequate | 15 | 9% |
| Scientific knowledge | 107 | 65% |
| Enforcement mechanisms | 75 | 45% |
| Legal expertise and advice | 35 | 21% |
| Collaboration amongst stake holders | 105 | 64% |
| Collaboration amongst governmental departments | 81 | 49% |
| Administration staff | 10 | 6% |
| Other, please specify | 20 | 12% |

**Q42.** How would you assess management of the fishery you are involved in?

| Question | Strongly Agree | Agree | Neutral | Disagree | Strongly Disagree | Total Responses |
|---|---|---|---|---|---|---|
| Current management is sufficient to ensure the long-term sustainability of fishery | 19 | 55 | 20 | 50 | 16 | 160 |
| There needs to be stricter regulations on commercial fishing | 25 | 46 | 29 | 50 | 9 | 159 |
| There needs to be stricter regulations on recreational fishing | 17 | 37 | 53 | 41 | 12 | 160 |
| Current commercial fishing regulations are adequately enforced | 14 | 53 | 29 | 49 | 17 | 162 |
| Current management is sufficient to ensure the long-term sustainability of overall biodiversity | 14 | 30 | 31 | 65 | 21 | 161 |
| There are too many regulations | 8 | 33 | 34 | 74 | 10 | 159 |
| The regulations are too complex to manage, monitor and measure successfully | 12 | 35 | 28 | 70 | 13 | 158 |

**Q43.** I would like to get some information on how satisfied you are with various aspects of your job. How satisfied are you with.

| Question | Very Satisfied | Satisfied | Neutral | Dissatisfied | Very Dissatisfied | Total Responses |
|---|---|---|---|---|---|---|
| Level of access you have to scientific fishing data | 27 | 80 | 17 | 37 | 4 | 165 |
| Number of other managers working with you | 11 | 54 | 61 | 29 | 1 | 156 |
| Resources to manage in the best way you know | 11 | 46 | 47 | 44 | 6 | 154 |
| Collaboration with scientists | 25 | 73 | 20 | 40 | 3 | 161 |
| Getting messages across to the decision makers | 7 | 37 | 28 | 70 | 20 | 162 |
| Decisions based on scientific expertise | 8 | 54 | 31 | 56 | 14 | 163 |
| Level of influence you have on decision making | 7 | 43 | 34 | 64 | 15 | 163 |
| Level of application of your work | 14 | 50 | 42 | 41 | 12 | 159 |

**Q44.** Do you believe that illegal, unreported and unregistered (IUU) fishing is a problem for your fishery?

| Answer | Response | % |
|---|---|---|
| Yes | 100 | 64% |
| No | 57 | 36% |
| Total | 157 | 100% |

**Q45.** How much of the total catch in your fishery do you believe is due to illegal, unreported and unregistered fishing?

| Answer | Response | % |
|---|---|---|
| None at all | 4 | 4% |
| Less than 5% | 11 | 11% |
| 6–15% | 20 | 21% |
| 16–30% | 21 | 22% |
| 31–40% | 14 | 15% |
| 41–50% | 15 | 16% |
| 51–60% | 6 | 6% |
| 61–80% | 0 | 0% |
| More than 80% | 5 | 5% |
| Total | 96 | 100% |

**Q46.** Do you believe that illegal, unreported and unregistered (IUU) fishing is a problem within your country?

| Answer | Response | % |
|---|---|---|
| Yes | 107 | 66% |
| No | 55 | 34% |
| Total | 162 | 100% |

**Q47.** How much of the total catch in your country do you believe is due to illegal, unreported and unregistered (IUU)?

| Answer | Response | % |
|---|---|---|
| None at all | 0 | 0% |
| Less than 5% | 7 | 7% |
| 6–15% | 23 | 22% |
| 16–30% | 39 | 38% |
| 31–40% | 13 | 13% |
| 41–50% | 13 | 13% |
| 51–60% | 3 | 3% |
| 61–80% | 3 | 3% |
| More than 80% | 3 | 3% |
| Total | 104 | 100% |

**Q48.** Do you believe that illegal, unreported and unregistered (IUU) fishing is a problem in some parts of the world?

| Answer | Response | % |
|---|---|---|
| Yes | 137 | 99% |
| No | 1 | 1% |
| Total | 138 | 100% |

**Q49.** How much of the total catch world-wide do you believe is due to illegal, unreported and unregistered (IUU)?

| Answer | Response | % |
|---|---|---|
| None at all | 0 | 0% |
| Less than 5% | 0 | 0% |
| 6–15% | 3 | 2% |
| 16–30% | 25 | 19% |
| 31–40% | 36 | 27% |
| 41–50% | 32 | 24% |
| 51–60% | 19 | 14% |
| 61–80% | 15 | 11% |
| More than 80% | 4 | 3% |
| Total | 134 | 100% |

**Q50.** What are the key aspects of these IUU problems?

| Answer | Response | % |
|---|---|---|
| Corruption | 80 | 66% |
| Lack of data | 53 | 44% |
| Poverty | 52 | 43% |
| No or little governance in place | 61 | 50% |
| No or little high seas controls | 52 | 43% |
| Lack of international policies | 34 | 28% |
| Lack of international compliance | 46 | 38% |
| Fishers' data not accurate | 57 | 47% |
| Growing human population | 34 | 28% |
| Lack of political will | 61 | 50% |
| Trawlers entering MPAs | 11 | 9% |
| High demand for high-valued fish species | 24 | 20% |

| Answer | Response | % |
|---|---|---|
| Recreational fishers | 11 | 9% |
| Large black market | 34 | 28% |
| Insufficient compliance | 67 | 55% |
| Not enough awareness of the consequences | 19 | 16% |
| Habitat destruction | 23 | 19% |
| Other | 7 | 6% |

**Q51.** What approaches does your organisation use to measure fish abundance?

| Answer | Response | % |
|---|---|---|
| No measures are used | 10 | 6% |
| Catch Per Unit Effort (CPUE) | 103 | 65% |
| Size | 75 | 47% |
| Recruitment | 58 | 36% |
| Fishers' log books | 100 | 63% |
| Government trawling data | 61 | 38% |
| Age structure | 66 | 42% |
| Other, please specify | 31 | 19% |

**Q52.** What improvements are needed to obtain/maintain sustainable fisheries?

| Answer | Response | % |
|---|---|---|
| No improvements are needed | 10 | 6% |
| Stronger political commitment to marine ecosystem management is needed | 119 | 72% |
| More regulation is needed | 38 | 23% |
| More science is needed | 88 | 53% |
| More enforcement is needed | 96 | 58% |
| Higher reliability and quality of catch data is needed | 85 | 52% |
| A higher level of ecosystem management is needed | 88 | 53% |
| Consumers drive the market and are responsible for buying sustainable seafood | 61 | 37% |
| Other | 12 | 15% |

**Q53.** How old are you?

| Answer | Response | % |
|---|---|---|
| 18–25 | 7 | 4% |
| 26–34 | 31 | 19% |
| 35–54 | 99 | 60% |
| 55–64 | 25 | 15% |
| 65 or over | 3 | 2% |
| Total | 165 | 100% |

**Q54.** What is the highest level of education you have completed?

| Answer | Response | % |
|---|---|---|
| Less than High School | 0 | 0% |
| High School/GED | 8 | 5% |
| Some College | 6 | 4% |
| 2-year College/University Degree | 8 | 5% |
| 3–4-year College/University Degree | 24 | 14% |

| Answer | Response | % |
|---|---|---|
| Master's Degree | 47 | 28% |
| Doctoral Degree | 71 | 42% |
| Professional Degree (JD, MD) | 4 | 2% |
| Total | 168 | 100% |

**Q55.** What is your degree in?

| Answer | Response | % |
|---|---|---|
| Marine science | 89 | 59% |
| Environmental science | 30 | 20% |
| Business and Management | 11 | 7% |
| Economics | 4 | 3% |
| Law | 4 | 3% |
| Political science | 5 | 3% |
| Social science | 5 | 3% |
| Other (please specify) | 10 | 7% |

**Q56.** What is your gender?

| Answer | Response | % |
|---|---|---|
| Female | 47 | 29% |
| Male | 117 | 71% |
| Total | 164 | 100% |

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
