# Peer review of "How to Sustain Fisheries: Expert Knowledge from 34 Nations"

_water, doi:10.3390/w11020213_

Round 1

Reviewer 1 Report

The manuscript provides useful insights for policy makers and manager of fisheries, while confirming that sustaining fisheries is a complex challenge. Sufficient reviews of existing literatures are provided. At the same time, to further improve the scientific quality of the manuscript, modifications for several points are recommended.

First, the purpose of the research is not adequately written. The current text states “this paper attempted to identify the main issues with sustaining fisheries” (line 121). But a research question is not clear. The author may wish to provide additional explanations on a research question on this study.

Second, this research argues that a lack of political-will is the main challenge, but the analysis mainly used the answers obtained from scientists. Underrepresentation of policy makers as respondents is apparent, as answers of Q4 indicate that policy makers occupy only 7% of the respondents while scientists are 54% of the respondents. Is there any potential bias for the research results caused by the selection of respondents that is overrepresented by scientists?

Third, the research only used data obtained in 2012. Are there any other attempts to obtain data using similar opportunities after 2012?

Lastly, further editorial checking is needed. For instance, incomplete sentence is found at line 101.

Author Response

First, the purpose of the research is not adequately written. The current text states “this paper attempted to identify the main issues with sustaining fisheries” (line 121). But a research question is not clear. The author may wish to provide additional explanations on a research question on this study.

Second, this research argues that a lack of political-will is the main challenge, but the analysis mainly used the answers obtained from scientists. Underrepresentation of policy makers as respondents is apparent, as answers of Q4 indicate that policy makers occupy only 7% of the respondents while scientists are 54% of the respondents. Is there any potential bias for the research results caused by the selection of respondents that is overrepresented by scientists?

Third, the research only used data obtained in 2012. Are there any other attempts to obtain data using similar opportunities after 2012?

Reviewer 2 Report

This is a valuable contribution.   There are some typos (title of Figure 1) and sentence fragments (line 101) to clear up.  IUU should be more clearly defined.   The conclusion could be more expansive, and the appendix is perhaps too long.   Perhaps the questions and details of the survey could be made available separately on request. 

Author Response

We thank the reviewers for their comments. We have read over the manuscript, directly addressing the comments as noted below, but also generally trying to improve readability over the paper.

Reviewer 1

First, the purpose of the research is not adequately written. The current text states “this paper attempted to identify the main issues with sustaining fisheries” (line 121). But a research question is not clear. The author may wish to provide additional explanations on a research question on this study.

-        We have rewritten the final section of the introduction in an attempt to make a clear statement on the intent of the work

Second, this research argues that a lack of political-will is the main challenge, but the analysis mainly used the answers obtained from scientists. Underrepresentation of policy makers as respondents is apparent, as answers of Q4 indicate that policy makers occupy only 7% of the respondents while scientists are 54% of the respondents. Is there any potential bias for the research results caused by the selection of respondents that is overrepresented by scientists?

-        Addressed, text added in a couple of places in the discussion to address this point.

Third, the research only used data obtained in 2012. Are there any other attempts to obtain data using similar opportunities after 2012?

-        We make a minor acknowledgement of this in the text and note what might be useful follow up. However, we have not personally been able to collect additional data since 2012 unfortunately.

Reviewer 2

This is a valuable contribution.   There are some typos (title of Figure 1) and sentence fragments (line 101) to clear up.  IUU should be more clearly defined.   The conclusion could be more expansive, and the appendix is perhaps too long.   Perhaps the questions and details of the survey could be made available separately on request.  

-        Have tried to find all typos and to make sure all acronyms are clearly defined (including IUU)

-        The fragment was a sub-heading and has now been fixed.

-        We have added a small amount of additional text to the Conclusion.

-        If WATER wants the Appendix separately (e.g. as an online supplement) that is fine by the authors; however, we will defer to the editor’s decision.
